# Angiopoietin-2–integrin α5β1 signaling enhances vascular fatty acid transport and prevents ectopic lipid-induced insulin resistance

Hosung Bae [1], Ki Yong Hong[2], Choong-kun Lee[1,3], Cholsoon Jang [4,8], Seung-Jun Lee[1], Kibaek Choe [5], Stefan Offermanns [6], Yulong He [7 ✉], Hyuek Jong Lee[1] & Gou Young Koh [1,3 ✉]

Proper storage of excessive dietary fat into subcutaneous adipose tissue (SAT) prevents ectopic lipid deposition-induced insulin resistance, yet the underlying mechanism remains unclear. Here, we identify angiopoietin-2 (Angpt2)–integrin α5β1 signaling as an inducer of fat uptake specifically in SAT. Adipocyte-specific deletion of Angpt2 markedly reduced fatty acid uptake and storage in SAT, leading to ectopic lipid accumulation in glucose-consuming organs including skeletal muscle and liver and to systemic insulin resistance. Mechanistically, Angpt2 activated integrin α5β1 signaling in the endothelium and triggered fatty acid transport via CD36 and FATP3 into SAT. Genetic or pharmacological inhibition of the endothelial integrin α5β1 recapitulated adipocyte-specific Angpt2 knockout phenotypes. Our findings demonstrate the critical roles of Angpt2–integrin α5β1 signaling in SAT endothelium in regulating whole-body fat distribution for metabolic health and highlight adipocyte–endothelial crosstalk as a potential target for prevention of ectopic lipid deposition-induced lipotoxicity and insulin resistance.

[1] Center for Vascular Research, Institute for Basic Science, Daejeon 34141, Republic of Korea. [2] Department of Plastic and Reconstructive Surgery, Dongguk University Ilsan Hospital, Goyang 10326, Republic of Korea. [3] Graduate School of Medical Science and Engineering, Korea Advanced Institute of Science and Technology (KAIST), Daejeon 34141, Republic of Korea. [4] Lewis Sigler Institute for Integrative Genomics and Department of Chemistry, Princeton University, Washington Rd, Princeton, NJ 08544, USA. [5] Graduate School of Nanoscience and Technology, Korea Advanced Institute of Science and Technology (KAIST), Daejeon 34141, Republic of Korea. [6] Department of Pharmacology, Max Planck Institute for Heart and Lung Research, 61231 Bad Nauheim, Germany. [7] Cyrus Tang Hematology Center, Collaborative Innovation Center of Hematology, Soochow University, 215123 Suzhou, China. [8] Present address: Department of Biological Chemistry, University of California Irvine, 92697 Irvine, CA, US. ✉email: heyulong@suda.edu.cn; gykoh@kaist.ac.kr

The adipose tissue plays a pivotal role in maintaining whole-body energy homeostasis by buffering lipids[1–3]. Impaired uptake of circulating lipids by subcutaneous adipose tissue (SAT) can induce ectopic fat accumulation in major glucose-consuming organs such as the skeletal muscle and liver[4,5], leading to lipotoxicity and insulin resistance[6,7]. Several approaches have focused on preventing ectopic fat accumulation; one strategy inhibits trans-endothelial fatty acid (FA) transport[8–11], because the vascular endothelial cell (EC) is the anatomic and metabolic gatekeeper of lipid shuttling into tissues[12,13]. However, approaches targeting trans-endothelial FA transport have been only partly successful[14,15], presumably because of the systematic approach targeting the whole circulatory system rather than specific tissues. Therefore, it will be clinically important to develop new ways to control tissue-specific endothelial FA transport, such as driving FA trafficking into SAT.

Transducing roles for vascular ECs in FA trafficking into tissues have been uncovered through the recent characterization of FA transport proteins (FATPs) and CD36 in ECs[12,16]. These proteins are regulated by autocrine or paracrine factors such as VEGF-B, apelin, or Notch ligands that act on receptors of ECs[8,10,11,17]. Other than its corresponding receptors such as VEGFR1, NRP1, or APLNR, integrin receptors are also implicated in FA handling[18,19], but their role in endothelial FA trafficking is unknown. Among the well-known ligands of integrin receptors are ECM molecules such as fibronectin, Mfge8, and Angpt2[20–22]. In particular, the effects of Angpt2 in regulating adipose tissue metabolism through angiogenesis have been thoroughly investigated[23,24]. However, approaches targeting adipose tissue angiogenesis through systemic blockade or constant overexpression have generated mixed results in metabolic outcome[25–28]. Thus, more thorough investigation is required to shed light on the fundamental mechanisms underlying adipocyte crosstalk with ECs by other methods such as adipocyte-specific deletion of Angpt2.

Here, using adipocyte-specific KO of Angpt2 and endothelial-specific KO of integrin receptors, we identify Angpt2–integrin α5β1 signaling as a novel regulator of trans-endothelial FA transport, specifically in subcutaneous adipose depots. Angpt2–integrin α5β1 induced FA transport into SAT is critical for clearance of circulating FAs and prevention of peripheral lipid accumulation and systemic glucose intolerance. Thus, stimulation of FA transport by targeting SAT endothelium through Angpt2–integrin α5β1 signaling offers a new therapeutic avenue for combat against ectopic lipid-induced insulin resistance and related metabolic syndrome.

## Results

### Adipocyte-derived Angpt2 leads subcutaneous fat distribution.
To validate the expression of Angpt2 in adipose tissues[23], we first examined the expressions and distributions of Angpt2 in various metabolic organs. Angpt2 expression was highest in SAT compared with other adipose tissues or metabolically active organs in mice (Supplementary Fig. 1a–c). In SAT, Angpt2 expression was enriched in adipocytes compared with stromal vascular fraction (SVF) (Supplementary Fig. 1d).

To investigate the role of adipocyte-derived Angpt2, we generated adipocyte-specific Angpt2 knockout (KO) mice (Angpt2$^{\Delta Ad}$) by crossing the adiponectin-Cre line with Angpt2$^{fl/fl}$ mice[29] and analyzed them at 8 weeks after birth (Supplementary Fig. 2a). Cre-negative but flox/flox-positive mice among the littermates were defined as wild-type (WT) mice for each experiment unless otherwise indicated. We found no differences in body weight (BW) and visceral adipose tissue (VAT) weights between WT and Angpt2$^{\Delta Ad}$ mice (Supplementary Fig. 2b, c).

However, we found altered fat distribution among different depots in Angpt2$^{\Delta Ad}$ mice, with significantly contracted SAT and enlarged brown adipose tissue (BAT) (Supplementary Fig. 2c, d). Inducible adipocyte-specific Angpt2 KO (Angpt2$^{i\Delta Ad}$) mice also showed almost identical phenotypes, indicating that this alteration was not due to developmental defects (Fig. 1a–d). Interestingly, the fat mass change was largely due to the altered size in adipocytes: the size of SAT adipocytes was reduced while the size of BAT adipocytes was increased (Fig. 1e). However, these smaller SAT adipocytes in the Angpt2$^{i\Delta Ad}$ mice did not show any signs of apoptosis, beiging, oxygen consumption, immune cell infiltration, or defective vascularization (Fig. 1f–k and Supplementary Fig. 3a–g), suggesting that the reduced size is due to decreased fat contents. Compared with WT mice, EC-specific Angpt2 KO mice (Angpt2$^{i\Delta EC}$) showed no differences in BW or weights and adipocyte sizes of SAT, VAT, and BAT (Fig. 1l–p). Thus, Angpt2 derived from the adipocytes but not from the ECs regulates the reciprocal distribution of fat into SAT and BAT.

### Angpt2 stimulates endothelial FA uptake.
We next sought to understand how fat contents were selectively reduced in SAT by Angpt2 deletion. Thus, we examined if Angpt2 affects FA trafficking into adipocytes by measuring tissue uptake of orally administered radio-labeled FAs to Angpt2$^{i\Delta Ad}$ mice (Fig. 2a). Angpt2$^{i\Delta Ad}$ mice showed 56% reduction in FA uptake by SAT, whereas they showed increased FA uptake by BAT (Fig. 2b and Supplementary Fig. 4a). Because BAT expresses much less Angpt2 than SAT (Supplementary Fig. 1a), we speculate that the increased FA uptake by BAT reflects compensation mechanisms of reduced FA uptake by SAT. Importantly, FA production, uptake by liver, or systemic levels of insulin or glucose, which are regulators of FA mobilization, were unchanged, indicating direct action of Angpt2 on FA uptake (Supplementary Fig. 4b–j).

To elucidate the mechanism of Angpt2 action on FA uptake, we measured FA intake in isolated SAT adipocytes in vitro (Fig. 2c). To our surprise, we found no difference in FA uptake between WT and Angpt2-deficient (Angpt2$^{\Delta Ad}$) adipocytes (Fig. 2d, e), indicating that Angpt2 induces fat uptake in a cell-nonautonomous manner. Given that blood vessels act as a gatekeeper of FA trafficking into tissues[12,30] and that ECs express Angpt2 receptors (e.g., Tie2 and integrins)[21], we tested if Angpt2 regulates endothelial FA transport. We found that Angpt2 treatment significantly increased FA uptake by ECs (Fig. 2f, g). These results indicate that Angpt2 acts as an adipokine to stimulate endothelial FA uptake in a cell-nonautonomous manner.

### Organotypic characteristics of endothelial cells in SAT.
To search for the receptor(s) that mediate the Angpt2 effect, we assessed which Angpt2 receptors are highly expressed in ECs of adipose tissues using the RiboTag$^{\Delta EC}$ mouse[31,32] to avoid disruption in cell surfaces. Upon tamoxifen treatment, this mouse tags hemagglutinin (HA) to the ribosome-associated actively transcribing mRNAs, specifically in VE-Cadherin-expressing ECs (Fig. 3a–c). After successful validation of EC-specific marker enrichment [e.g., platelet endothelial cell adhesion molecule (PECAM)] in mRNAs purified with an antibody against the HA tag (Fig. 3d), we compared a catalog of known Angpt2 receptors in ECs from various organs (Fig. 3e). Of note, the ECs in SAT expressed higher levels of integrin α5 (ITGα5) and integrin β1 (ITGβ1) by ~5.6-fold but ~52% reduced levels of Tie2 compared with ECs in the other organs we examined (Fig. 3e). Immuno-histochemical analyses supported these reciprocal expression levels of integrin α5β1 and Tie2 in the ECs of SAT, which clearly differs with other depots (Fig. 3f, g and Supplementary Fig. 5a, b).

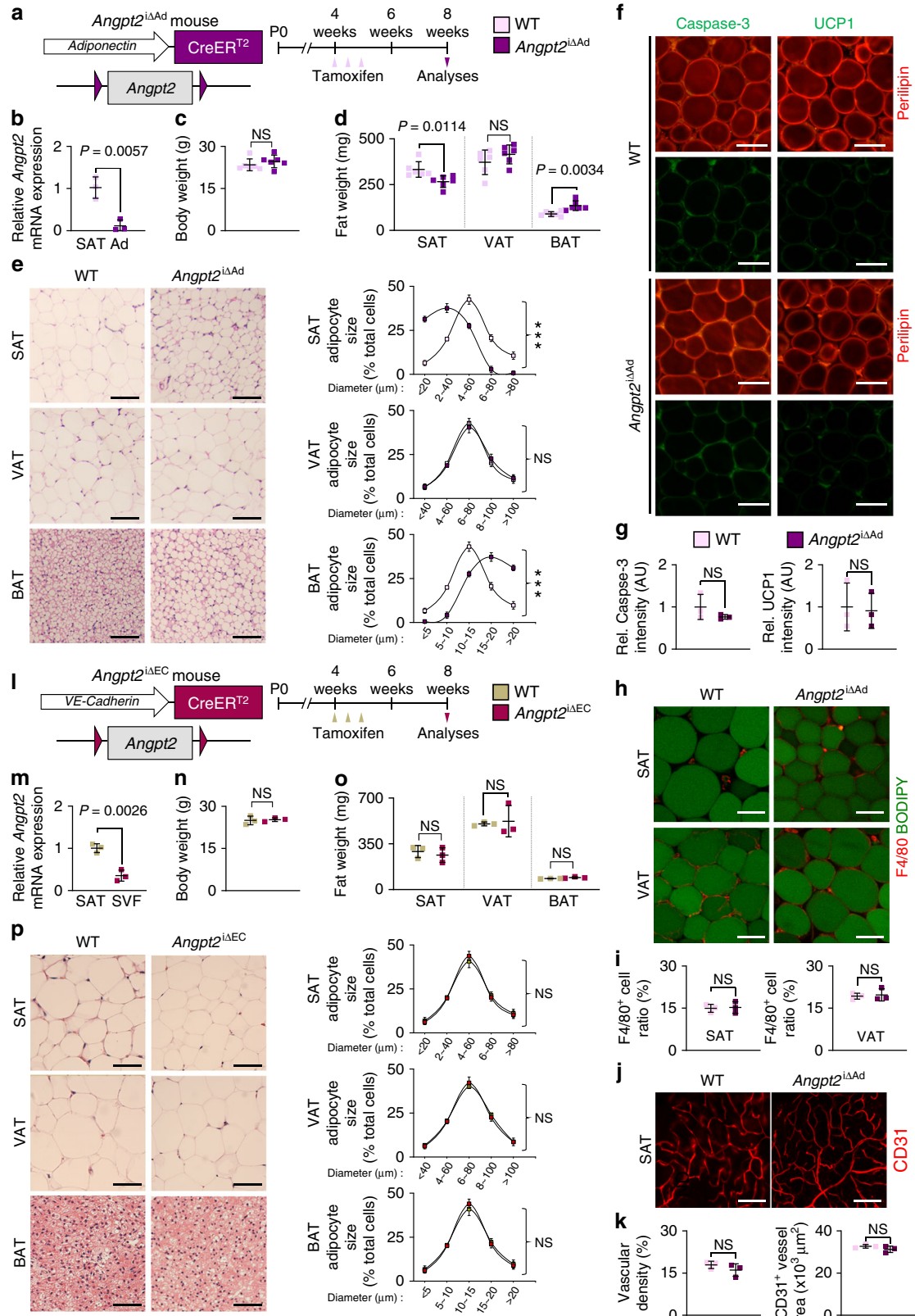

Thus, integrins are the strongest candidate receptors that are likely mediators of Angpt2 action on the endothelium in SAT.

**Angpt2 drives endothelial FA uptake through integrin α5β1.** To test this hypothesis, we used uncoated plates to minimize integrin activation by ECM proteins such as gelatin or fibronectin[33]; nonetheless, the adherence of HUVECs was comparable to coated dishes (Supplementary Fig. 6a). Next, we examined the effect of $Mn^{2+}$, which enhances ligand-binding affinity of integrin[21], on endothelial FA uptake (Fig. 4a). Treatment of $Mn^{2+}$ alone showed no effect, but it strongly augmented Angpt2-induced FA uptake by ECs by 3.0-fold (Fig. 4b, c). In the presence of $Mn^{2+}$, Angpt2 stimulated FA uptake rapidly (Fig. 4d)

**Fig. 1 Depletion of Angpt2 from adipocytes alters subcutaneous fat distribution. a** Diagram for generation of $Angpt2^{i\Delta Ad}$ mice, inducible and specific deletion of Angpt2 in adipocytes by tamoxifen delivery into 4 week old mice and analyses in 8-week old mice. **b** Comparisons of Angpt2 mRNA expression in fractionized adipocytes (Ad) of SAT in WT and $Angpt2^{i\Delta Ad}$ mice. **c, d** Comparison of body weight and fat weight in different adipose tissues between WT and $Angpt2^{i\Delta Ad}$ mice. $n = 6$ mice/group pooled from three independent experiments. **e** Representative H&E-stained images and comparisons in adipocyte size (diameter; μm) of different adipose tissues in WT and $Angpt2^{\Delta Ad}$ mice. Four to six different mice of each genotype were randomly selected to determine the adipocyte size and data are presented as % of total cells. Scale bars, 50 μm. **f, g** Representative images and comparisons of apoptosis (Caspase-3) and beiging (UCP1) in SAT of WT and $Angpt2^{i\Delta Ad}$ mice. Scale bars, 50 μm. **h, i** Representative images and comparison of indicated immune cell infiltration in SAT and VAT of WT and $Angpt2^{i\Delta Ad}$ mice. Magnified view is shown in right panels. Scale bars, 30 μm. **j, k** Representative images and comparisons of vascular density and CD31+ vessel area in SAT of WT and $Angpt2^{i\Delta Ad}$ mice. Scale bars, 100 μm. **l** Diagram for generation of $Angpt2^{i\Delta EC}$ mice, inducible and specific deletion of Angpt2 in endothelial cells by tamoxifen delivery into 4 week old mice and analyses in 8 week old mice. **m** Comparisons of Angpt2 mRNA expression in stromal vascular fraction (SVF) of SAT in WT and $Angpt2^{i\Delta EC}$ mice. **n, o** Comparison of body weight and fat weight in different adipose tissues between WT and $Angpt2^{i\Delta EC}$ mice. **p** Representative H&E-stained images and comparisons in adipocyte size (diameter; μm) of different adipose tissues in WT and $Angpt2^{i\Delta EC}$ mice. Four different mice of each genotype were randomly selected to determine the adipocyte size and data are presented as % of total cells. Scale bars, 50 μm. Unless otherwise denoted, each dot indicates a value obtained from one mouse and $n = 3$ mice/group pooled from two independent experiments. Horizontal bars indicate mean ± SD and P values versus WT by two-tailed Student's t test. NS not significant.

and dose dependently (Fig. 4e). We found no difference between vascular leakage by trans-well endothelial layer permeability assay, indicating that Angpt2 stimulates FA uptake independently of vascular permeabilizing actions (Fig. 4f). We next depleted various integrin subunits in ECs to further examine the importance of each integrin in mediating the Angpt2 effect. Depletion of integrin α5 or β1 completely blocked Angpt2-induced FA uptake by ECs (Fig. 4g), whereas depletion of integrin αv or β5 did not (Supplementary Fig. 6b). Of interest, depletion of Tie2 or integrin β3 rather enhanced Angpt2-induced FA uptake (Supplementary Fig. 6b), presumably because the absence of these receptors liberates Angpt2 and allows more Angpt2 to activate integrin α5β1[21,34]. Consistent with the depletion data, the α5β1-specific blocking peptide ATN-161[35] completely suppressed Angpt2-induced FA uptake (Fig. 4h). Conversely, the conformational activator of integrin α5β1, SNAKA-51[22,36], enhanced Angpt2 activity (Fig. 4i). We observed similar effects of Angpt2 in a mouse EC cell line (MS1) but not in adipocytes isolated from SAT (Supplementary Fig. 6c–e).

To strengthen our finding that Angpt2 induces organotypic FA uptake in SAT ECs, we compared the effect of Angpt2 on primary ECs from SAT and VAT (Supplementary Fig. 7a). First, we employed a previously published method for culturing primary ECs of murine organs[37], and validated its 92.7% purity (Supplementary Fig. 7a–c). Next, we compared the effects of Angpt2 treatment with or without $Mn^{2+}$ in primary ECs from SAT and VAT (Supplementary Fig. 7d). Of note, Angpt2 treatment alone enhanced FA uptake in time- and dose-dependent manners only in SAT ECs (Supplementary Fig. 7d±f). Importantly, this effect was inhibited by ATN-161 treatment (Supplementary Fig. 7g). These data demonstrate that the endothelial integrin α5β1 in SAT mediates Angpt2-induced FA uptake.

**Angpt2–integrin α5β1 drives FA transport through CD36/FATP3.** Various FATPs mediate endothelial FA uptake[12,30]. Of note, Angpt2-induced FA uptake was specific for the long-chain FAs (Fig. 5a). We thus depleted candidate FA transporters in ECs, including FA translocase (CD36) and FATPs (Fig. 5b). Also of interest, depletion of CD36 or FATP3, but not of FATP4, blocked Angpt2-induced FA uptake and transport by ECs (Fig. 5c–f). However, we found no changes in gene expression levels of CD36 or FATP3 after Angpt2 treatment (Fig. 5g). Thus, Angpt2 activates endothelial FA uptake, likely via redistribution or protein–protein interactions of CD36 or FATP3[9].

Intracellular translocation of CD36 or FATPs, and consequently increased FA uptake, have been reported in various cell types[16,20]. Therefore, we tracked protein expression of CD36 or

FATP3 in ECs after Angpt2 treatment. Although we did not observe any changes in localization of FATP3 in ECs (Fig. 5h), we detected rapid formation of punctate CD36 structures in perinuclear regions after only 5 min following Angpt2 addition (Fig. 5i). Intriguingly, these punctate CD36 signals were co-localized with ITGβ1 (Fig. 5i), suggesting that CD36 and ITGβ1 may physically interact upon activation by Angpt2. To test this possibility, we conducted an in situ proximity ligation assay and observed strong signals indicative of complex formation between CD36 and ITGβ1 only after Angpt2 treatment (Fig. 5j, k). Moreover, ATN-161 markedly blunted CD36 co-localization with ITGβ1 and their interaction (Fig. 5l–n), indicating that activation of ITGβ1 by Angpt2 binding is required for CD36 translocation and interaction with ITGβ1. Immunoprecipitation (IP) by anti-CD36 antibody also demonstrated that CD36 bound to ITGβ1 and that their interaction increased by ~2.1-fold after Angpt2 treatment (Fig. 5o, p). Thus, Angpt2 facilitates CD36 translocation through integrin α5β1.

Since ITGβ1 faces the basolateral side of ECs[38], and CD36 is located on the luminal side of ECs[39], we speculated that transmembrane lipid rafts, which interacts with both molecules[40], could mediate this binding complex. Indeed, we detected co-localization of this complex in Caveolin-1+ lipid rafts in plasma membrane of ECs (Supplementary Fig. 8a, b). In line with this finding, we found that integrin α5β1 is mainly located on collagen IV+ CD34− basolateral membrane of ECs[41,42] in SAT (Supplementary Fig. 8c, d).

**Endothelial ITGβ1 stimulates FA transport into SAT.** We next proceeded to recapitulate our findings in vivo. First, we generated EC-specific inducible ITGβ1 KO ($ITG\beta1^{\Delta EC}$) mice by crossing VE-Cadherin-Cre-$ER^{T2}$ mice and $Itgb1^{flox/flox}$ mice, and analyzed them 1 week after tamoxifen administration due to lethality[43] (Fig. 6a). There were no changes in vascular integrity or density in $ITG\beta1^{\Delta EC}$ mice after 1 week of tamoxifen treatment (Supplementary Fig. 9a–d). Compared with WT mice, adipocyte sizes in SAT decreased by 63% and SAT weight by 23%, but no differences were found in VAT in $ITG\beta1^{\Delta EC}$ mice (Fig. 6b). We found no difference in BW, liver and muscle fat content, or BAT weight (Supplementary Fig. 9e–g). $ITG\beta1^{\Delta EC}$ mice also showed 42% reduced uptake of intravenously administered FAs toward SAT but not by VAT, BAT, or liver (Fig. 6c–e and Supplementary Fig. 9h, i). Thus, endothelial ITGβ1 is required for vascular FA transport toward SAT.

**Inhibition of integrin α5β1 blocks FA transport into SAT.** We next administered the integrin α5β1-specific blocking peptide

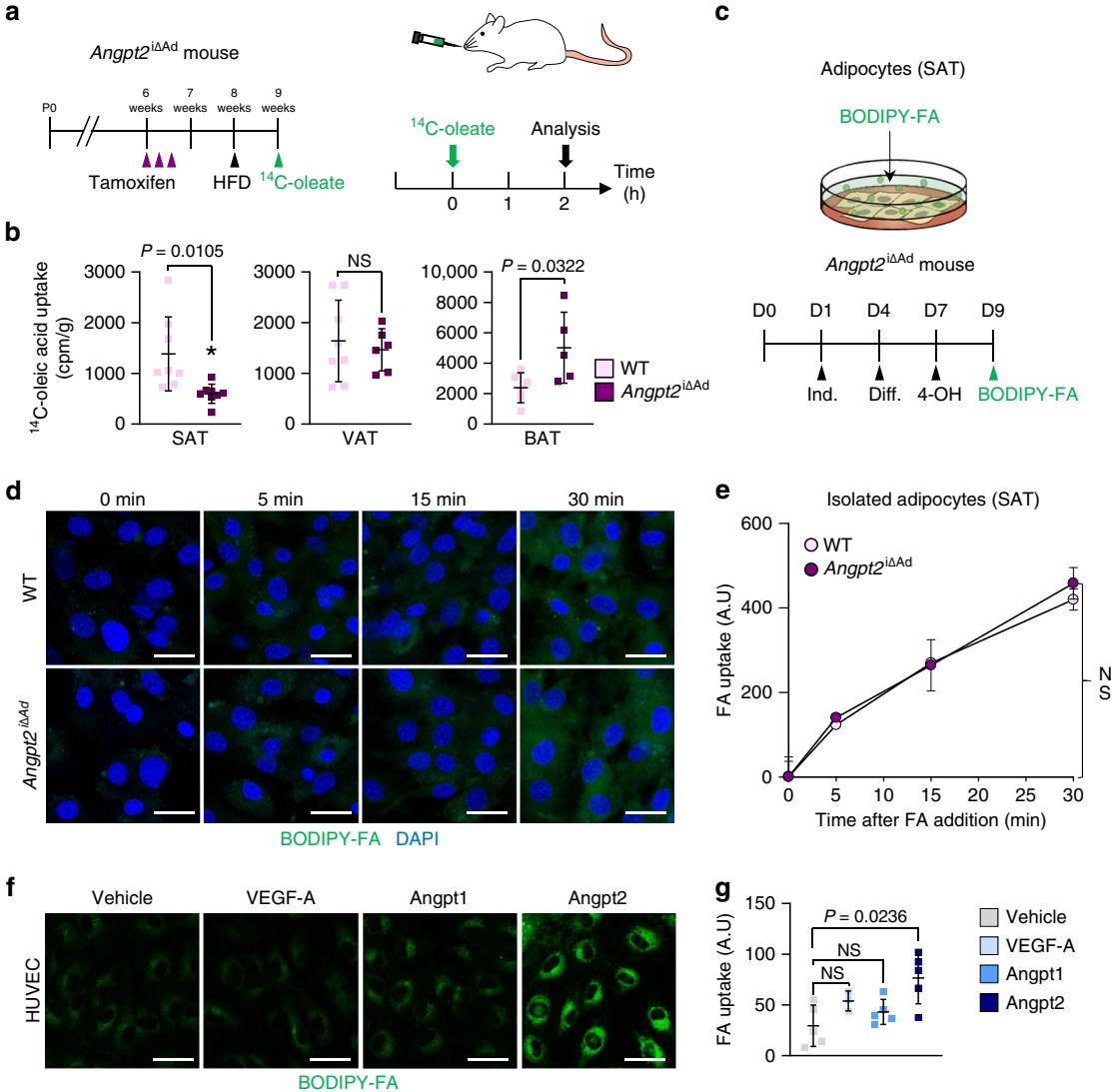

**Fig. 2 Angpt2 stimulates endothelial FA uptake. a** Diagram depicting the experiment scheme for examining fatty acid ($^{14}$C-oleate) transport into different adipose tissues in *Angpt2*$^{iΔAd}$ mice delivered with tamoxifen into 6 week old mice and HFD from 8 week old mice and their analyses in 9 week old mice. Adipose tissues were harvested 2 h after oral gavage of 2 mCi of $^{14}$C-oleate dissolved in 200 μl olive oil. **b** Comparisons of $^{14}$C-oleic acid uptake in different adipose tissues in WT and *Angpt2*$^{iΔAd}$ mice. Each dot indicates a value obtained from one mouse using *n* = 8(SAT), 8(VAT WT), 6(VAT *Angpt2*$^{iΔAd}$), 6 (BAT WT), and 5(BAT *Angpt2*$^{iΔAd}$) mice/group pooled from two independent experiments. Horizontal bars indicate mean ± SD and *P* values versus WT by two-tailed Student's *t* test. NS not significant. **c** Diagram depicting FA uptake of primary cultured adipocytes between WT and *Angpt2*$^{iΔAd}$ mice. SVF of SAT from WT and *Angpt2*$^{iΔAd}$ mice were isolated and cultured until confluence with adipocyte induction and differentiation medium at indicated days. After 2 days of 4-OH treatment to delete *Angpt2*, cells were analyzed for FA uptake. **d**, **e** Representative images and comparisons of FA (BODIPY C-12, 0.75 μM) uptake at indicated time points in primary cultured adipocytes from SAT of WT and *Angpt2*$^{iΔAd}$ mice. Each dot indicates a value obtained from one mouse using *n* = 3 mice/group pooled from two independent experiments. Horizontal bars indicate mean ± SEM. NS not significant. Scale bars, 30 μm. **f**, **g** Representative images and comparisons of FA (BODIPY C-12, 8 μM) uptake in HUVECs after treatment with VEGF-A (40 ng/ml), Angpt1 (200 ng/ml), or Angpt2 (2.5 μg/ml) for 30 min. *n* = 3 (VEGF-A), 5(Vehicle, Angpt1, Angpt2) for each group. Horizontal bars indicate mean ± SD and *P* values versus vehicle by one-way ANOVA followed by Tukey's multiple comparison test. Scale bars, 30 μm.

ATN-161 to WT mice (Supplementary Fig. 10a) to evaluate whether the phenotypes of *ITGβ1*$^{ΔEC}$ mice were caused by integrin α5β1 inhibition (Fig. 7a). Indeed, ATN-161 treatment recapitulated the *ITGβ1*$^{ΔEC}$ mouse SAT phenotypes, and the phenotype was even more enhanced after an additional week of treatment (Fig. 7b, c). This change was accompanied by a markedly diminished uptake of radioactive FAs in SAT, by 44%, but not by other organs (Fig. 7d). Again, we found no changes in BW, liver, and BAT weights (Supplementary Fig. 10b–e). Therefore, we concluded that the endothelial integrin α5β1 is essential for vascular FA transport, specifically in SAT.

**Angpt2–integrin α5β1 is enriched in nondiabetic obese SAT.** To investigate the clinical relevance of our findings, we compared gene expression profiles of SAT and VAT between nondiabetic obese (NDO) and diabetic-obese (DO) individuals from the publicly available gene expression database, Gene Expression Omnibus (GEO; GSE20950, GSE29226, GSE29231, GSE16415, GSE71416). Among the 34 genes that were specifically upregulated in SAT of NDO individuals, we identified Angpt2 as the sole secretory molecule (Fig. 8a and Supplementary Tables 1, 2).

To examine if dietary fat overload and obesity affect Angpt2 expression, we fed mice with high-fat diet (HFD). Surprisingly,

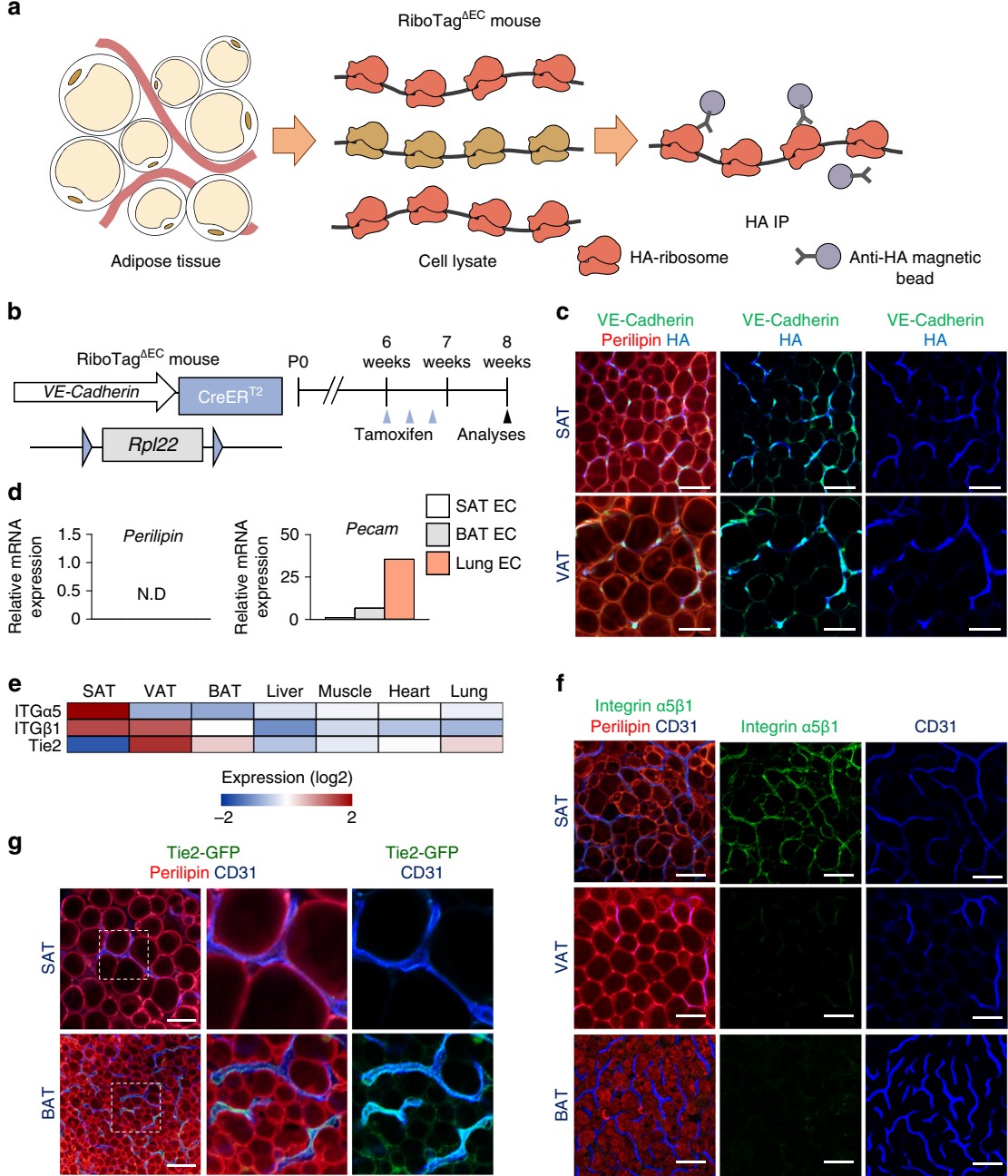

**Fig. 3 ECs of SAT are enriched with integrin α5β1. a** Diagram depicting EC-specific RiboTag transcriptome analysis. Inducible recombination of the RiboTag allele led to expression of hemaglutanin (HA)-tagged ribosomal protein (reddish orange) specifically in ECs. Ribosome-bound transcripts were immunoprecipitated (IP) from homogenized whole tissues with anti-HA antibody-coupled magnetic beads. Extracted mRNAs were analyzed by RNA-Seq analysis. **b** Generation of RiboTag$^{\Delta EC}$ mouse, inducible and specific tagging of ribosomes in ECs in 6-week-old mice, and their analyses 2 weeks later. Red arrowheads indicate daily injections of tamoxifen. **c** Representative images of HA-tagged ribosomal protein in VE-Cadherin expressing ECs in adipose tissues of RiboTag$^{\Delta EC}$ mouse. Scale bars, 50 μm. **d** Comparisons of *perilipin* and *pecam* expression in isolated mRNA of ECs from different organs of RiboTag$^{\Delta EC}$ mouse. **e** RNA-seq expression heatmap of ITGα5, ITGβ1, and Tie2 in isolated ECs from different organs using RiboTag$^{\Delta EC}$ mouse. $n = 5$ for each group. **f** Representative images of ITGα5β1 expression in various adipose tissues of WT mice. Scale bars, 50 μm. **g** Representative images of Tie2 expression in SAT and BAT of Tie2-GFP mice. Magnified views of dotted-line box are shown in lower panels. Scale bars, 50 μm.

with HFD, a gradual increase in Angpt2 expression was detected in SAT but not in other organs or in the systemic circulation (Fig. 8b, c and Supplementary Fig. 11a, b). Intriguingly, the Angpt2 induction was specific to the adipocytes (Fig. 8d and Supplementary Fig. 11c). This result was consistent with increased expression of Angpt2 in the isolated adipocytes from the SAT of NDO individuals (Fig. 8e). In detail, saturated FA (palmitic acid) treatment alone for 24 h was sufficient to increase

Angpt2 expression by 2.0- and 2.7-fold in SAT adipocytes in both mice and humans, respectively (Fig. 8f). Thus, Angpt2 expression is rapidly induced by dietary fat intake in a SAT adipocyte-specific manner.

By comparison analysis of human GEO data, we confirmed enhanced expression of ITGα5 in SAT compared with VAT in NDO patients (Fig. 8g). Likewise, we confirmed enriched expression of integrin α5β1 in ECs of SAT of NDO patient (Supplementary

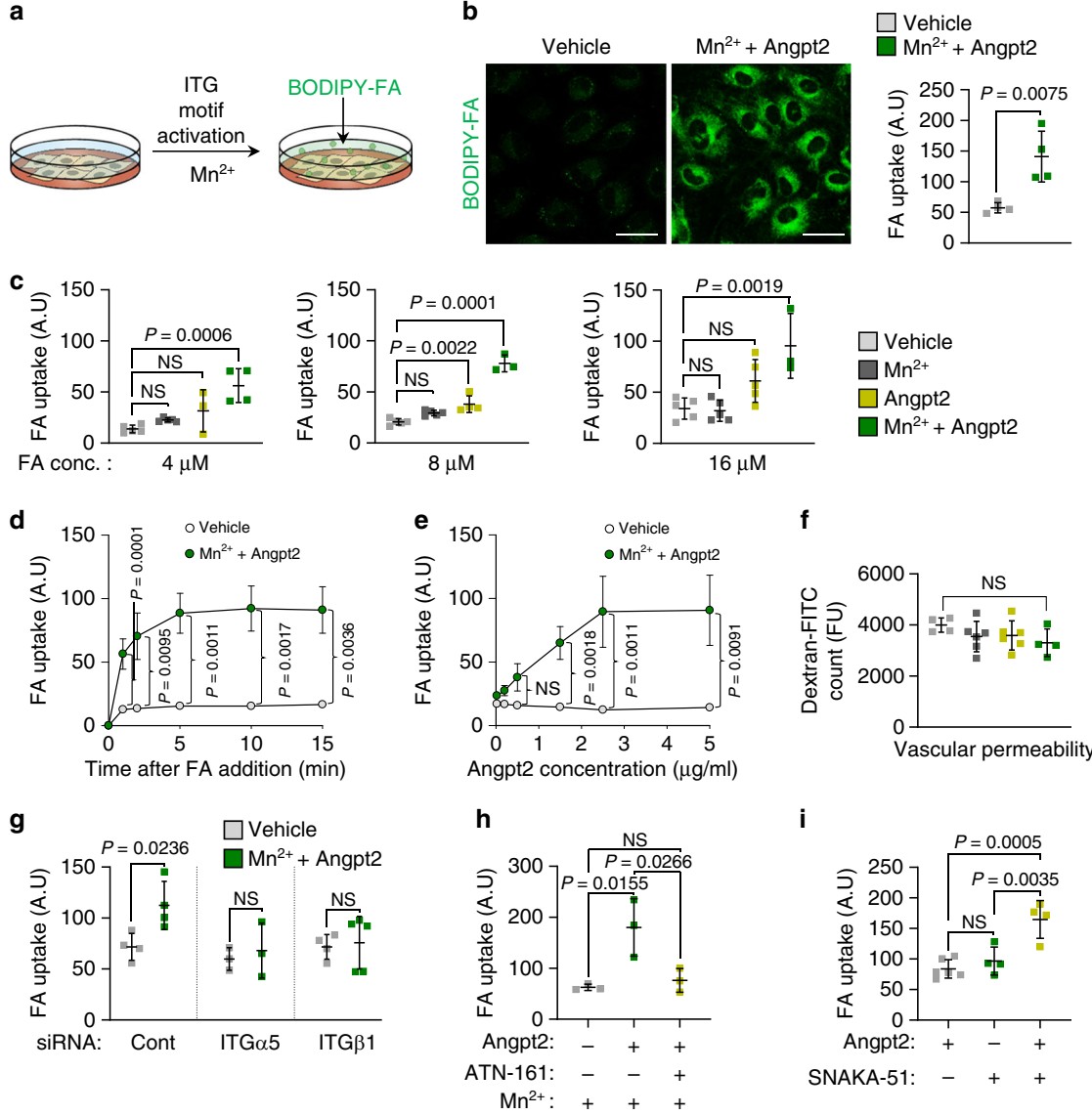

**Fig. 4 Angpt2 stimulates endothelial FA uptake through integrin α5β1 signaling. a** HUVECs were integrin-motif activated using 1 mM $Mn^{2+}$ and analyzed for FA (BODIPY C-12, 8 μM) uptake. **b** Representative images and comparison of FA uptake with vehicle or Angpt2 (2.5 μg/ml) treatment for 30 min in HUVECs with $Mn^{2+}$ (1 mM). Each dot indicates a mean of quadruplicate values from three independent experiments. Scale bars, 30 μm. **c** Comparisons of FA uptake at different concentrations with vehicle ($n = 5$), $Mn^{2+}$ (1 mM; $n = 5$), Angpt2 (2.5 μg/ml; $n = 3$ for 4 μM, 4 for 8 μM, 5 for 16 μM) or $Mn^{2+}$ (1 mM) + Angpt2 (2.5 μg/ml) ($n = 5$ for 4 μM; 3 for 8, 16 μM) treatment for 15 min in HUVECs. **d** Comparisons of FA uptake with vehicle ($n = 5$) or $Mn^{2+}$ (1 mM) + Angpt2 (2.5 μg/ml) ($n = 3$ for 1 min; 4 for 2, 5, 10 min; 5 for 15 min) treatment for indicated time points in HUVECs. **e** Comparisons of FA uptake treatment with vehicle ($n = 5$) or $Mn^{2+}$ (1 mM) + Angpt2 ($n = 3$ for 20 ng, 1.25, 2.5, 5 μg/ml; 4 for 200, 500 ng/ml) treatment with indicated concentration for 15 min in HUVECs. **f** Comparisons of vascular permeability with vehicle ($n = 4$), $Mn^{2+}$ (1 mM; $n = 6$), Angpt2 (2.5 μg/ml; $n = 6$) or $Mn^{2+}$ (1 mM) + Angpt2 (2.5 μg/ml) ($n = 4$) treatment for 15 min in HUVECs cultured in trans-well inserts. **g** Comparisons of FA uptake with vehicle or $Mn^{2+}$ (1 mM) + Angpt2 (2.5 μg/ml) treatment for 15 min in siControl ($n = 4$), siITGα5 ($n = 3$) or siITGβ1 ($n = 4$ for vehicle, 5 for $Mn^{2+}$ + Angpt2) HUVECs. **h** Comparisons of FA uptake with or without Angpt2 (2.5 μg/ml), ITGα5β1 blocking peptide (ATN-161, 10 μM) or $Mn^{2+}$ (1 mM) treatment for 15 min in HUVECs. Each dot indicates a mean of triplicate values from two independent experiments. **i** Comparisons of FA uptake with ($n = 6$) or without Angpt2 (2.5 μg/ml; $n = 4$) or ITGα5β1-specific motif activator (SNAKA-51, 10 μg/ml; $n = 4$) treatment for 15 min in HUVECs. Unless otherwise denoted, FA BODIPY C-12 concentration is 8 μM, horizontal bars indicate mean ± SD (**b**, **c**, **f–i**) or SEM (**d**, **e**) and P values versus vehicle by two-tailed Student's t test (**b**, **d**, **e**, **g**) or one-way ANOVA followed by Tukey's multiple comparison test (**c**, **f**, **h**, **i**). NS not significant.

Fig. 11d). Thus, integrin receptor expression is enriched in ECs of SAT in NDO individuals. Next, we observed changes in Angpt2 and its receptors during fast/fed cycle. Of note, Angpt2 expression was reduced by 58.8% in fasted mice (Supplementary Fig. 11e). Consistently, expressions of both ITGα5β1 and CD36 were downregulated in ECs during fasting (Supplementary Fig. 11f, g). These indicate that expressions of Angpt2 and its mediators in FA transport are reduced during energy output.

**Angpt2 prevents ectopic lipid-induced insulin resistance.** Given that Angpt2 in adipocytes is highly induced by dietary fat intake, we next challenged $Angpt2^{iΔAd}$ mice with HFD (Fig. 9a and Supplementary Fig. 12a). After 8 weeks of HFD, SAT weighed 28% less, and BAT weighed 71% more in $Angpt2^{iΔAd}$ mice compared with WT animals (Fig. 9b). Obesity-associated inflammatory markers in VAT were upregulated, while thermogenic markers were downregulated in BAT in $Angpt2^{iΔAd}$ mice (Supplementary Fig. 12b, c).

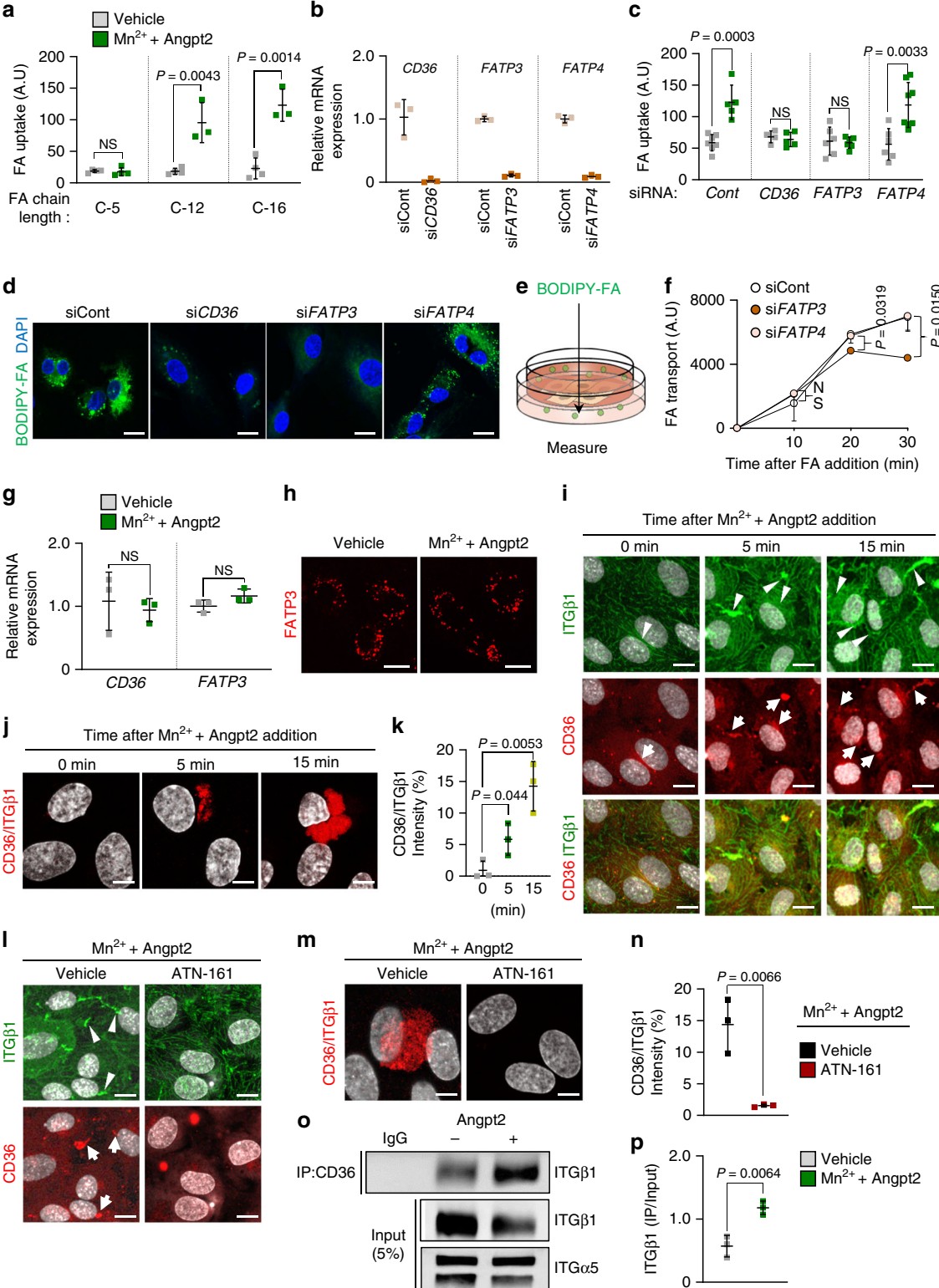

Likewise, electron microscope analysis revealed unpacked cristae and vacuole-filled mitochondria in BAT of *Angpt2*[iΔΔAd] mice (Supplementary Fig. 12d). Moreover, circulating triglyceride and leptin levels were each ~1.5-fold and ~2.0-fold higher, whereas plasma adiponectin level was 30% less in *Angpt2*[iΔΔAd] mice (Fig. 9c and Supplementary Fig. 12e, f). Of special note, histological analyses revealed profound lipid accumulation in the liver and skeletal muscle of *Angpt2*[iΔΔAd] (Fig. 9d, e). *Angpt2*[iΔΔAd] mice also showed

systemic glucose intolerance and insulin resistance (Fig. 9f, g), presumably because of ectopic fat deposition in these glucose-consuming organs[6,8]. Moreover, *Angpt2*[iΔΔAd] mice showed decreased metabolic rate without affecting food intake or activity (Fig. 9h, i and Supplementary Fig. 13a–e). Together, these data indicate that adipocyte-derived Angpt2 is required for proper fat distribution toward SAT to prevent lipid overflow into glucose-metabolizing organs during high-fat intake.

**Fig. 5 Angpt2–ITGα5β1 signaling facilitates FA transport through CD36 and FATP3. a**, **c**, **d**, **g**–**p** HUVECs were treated with vehicle or $Mn^{2+}$ (1 mM) + Angpt2 (2.5 µg/ml) for 15 min or indicated time points. **a** Comparisons of short-chain FA (BODIPY C-5, 8 µM; $n = 4$) and long-chain FA (BODIPY C-12 and C-16, 8 µM; $n = 4$ for vehicle and 3 for $Mn^{2+}$ + Angpt2) uptake. **b** Comparisons of depletion efficiency of *CD36*, *FATP3*, and *FATP4* in corresponding siRNA treated HUVECs. $n = 3$. **c**, **d** Comparisons and representative images of FA (BODIPY C-12, 8 µM) uptake in siControl ($n = 7, 5$), si*CD36* ($n = 4, 5$), si*FATP3* ($n = 6$), or si*FATP4* ($n = 6, 8$) HUVECs. Scale bars, 30 µm. **e** Diagram depicting the endothelial FA transport assay. HUVECs were cultured until confluence on trans-wells and FA (BODIPY C-12, 8 µM) was added to upper layer of trans-well, followed by analysis of transported FA in bottom well. **f** Comparisons of FA transport for indicated time points after $Mn^{2+}$ (1 mM) + Angpt2 (2.5 µg/ml) treatment in siControl ($n = 3$ for 10, 30 min and 4 for 20 min), si*FATP3* ($n = 3$) or si*FATP4* ($n = 4$) HUVECs. $n = 3$–5. **g** Comparison of mRNA expression of *CD36* and *FATP3*. **h** Representative images of FATP3-td-Tomato transfected HUVECs. Scale bars, 50 µm. **i** Representative images of CD36 and ITGβ1 expression. Expression of active ITGβ1 (arrows) and CD36 (arrowheads) are co-localized after Angpt2 treatment. Scale bars, 50 µm. **j**, **k** Representative images and comparison of in situ proximity ligation assay between CD36 and ITGβ1 complex. Scale bars, 20 µm. **l**–**n** Representative images and comparison of in situ proximity ligation assay between CD36 and ITGβ1 complex in HUVECs pre-treated with vehicle or ITGα5β1 blocking peptide (ATN-161, 10 µM) for 15 min. Scale bars, 50 µm (**l**); 20 µm (**m**). **o**, **p** Immunoprecipitation with anti-IgG and anti-CD36 antibody in HUVECs. Immunoblot analysis with anti-ITGβ1 and anti-ITGα5 antibodies are shown. Graph indicates normalized ratio of immunoprecipitated ITGβ1 per input. Unless otherwise denoted, each dot indicates a mean of triplicate values from three independent experiments and horizontal bars indicate mean ± SD or SEM (**f**) and *P* values versus control by two-tailed Student's *t* test. NS not significant.

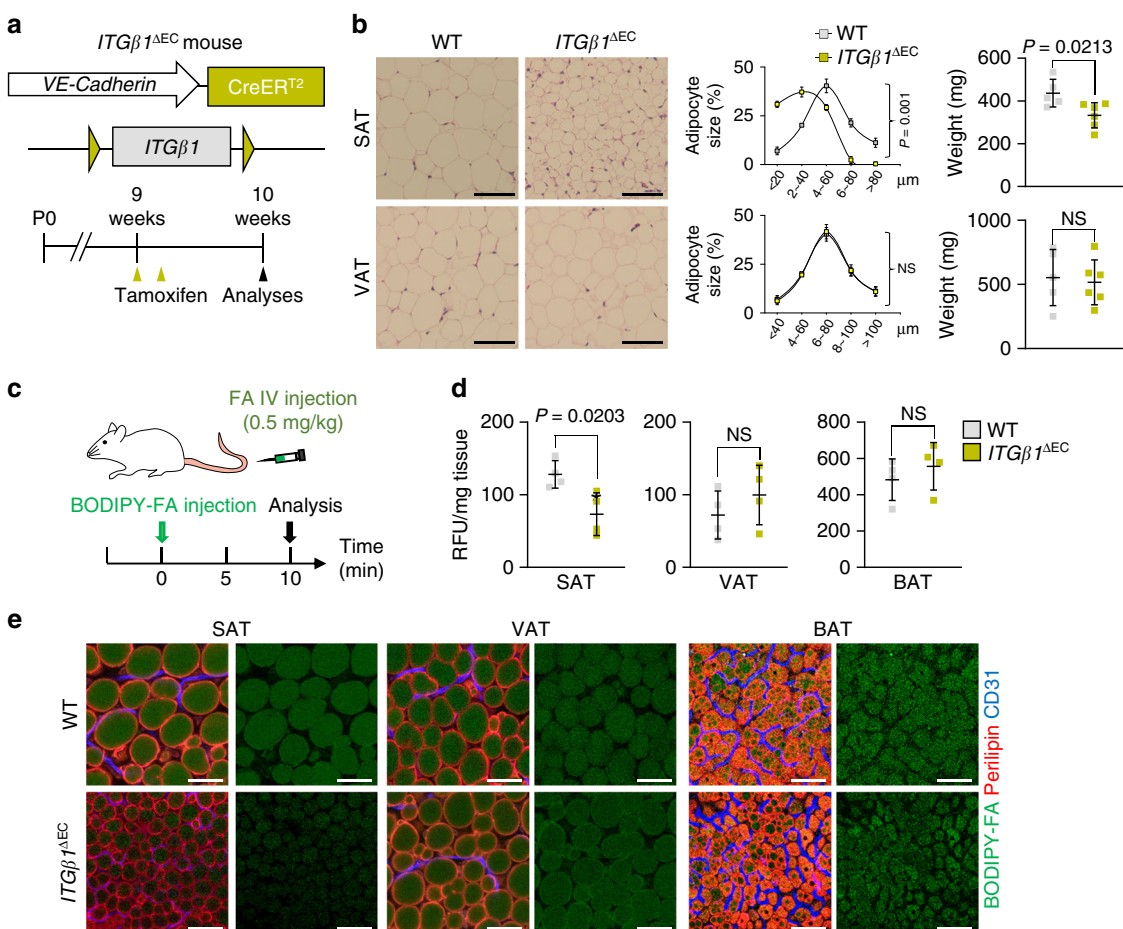

**Fig. 6 Endothelial ITGβ1 drives subcutaneous fat distribution. a** Generation of *ITGβ1*$^{ΔEC}$ mouse, inducible and specific deletion of *ITGβ1* in ECs in 9-week-old mice, and their analyses 1 week later. Yellow arrowheads indicate daily injections of tamoxifen. **b** Representative H&E-stained images and comparisons in adipocyte size (diameter; µm) of SAT and VAT in *ITGβ1*$^{ΔEC}$ mice. Four different mice of each genotype were randomly selected to determine the adipocyte size and data are presented as % of total cells. Horizontal bars indicate mean ± SD and *P* values versus WT by two-tailed Student's *t* test. Scale bars, 30 µm. **c** Diagram for intravenous FA supply in *ITGβ1*$^{ΔEC}$ mice. FA (BODIPY C-16, 0.5 g/kg) conjugated to BSA were administered intravenously into WT and *ITGβ1*$^{ΔEC}$ mice and analyzed 10 min after. **d**, **e** Comparisons and representative images of FA uptake into different adipose tissues between WT and *ITGβ1*$^{ΔEC}$ mice. Each dot indicates a mean of quadruplicate values from three independent experiments. Horizontal bars indicate mean ± SD and *P* value versus vehicle by two-tailed Student's *t* test. NS not significant. Scale bars, 30 µm.

## Discussion

Maintenance of metabolic health is of paramount importance in a society where obesity is rapidly on the rise. Our various tissue-specific KO mouse models and mechanistic studies in primary cultured cells demonstrate that adipocyte-produced Angpt2 regulates endothelial FA uptake via CD36 and FATP3 through integrin α5β1 signaling. Intriguingly, this process is critical for FA uptake specifically in subcutaneous fat depots. Inhibition of this

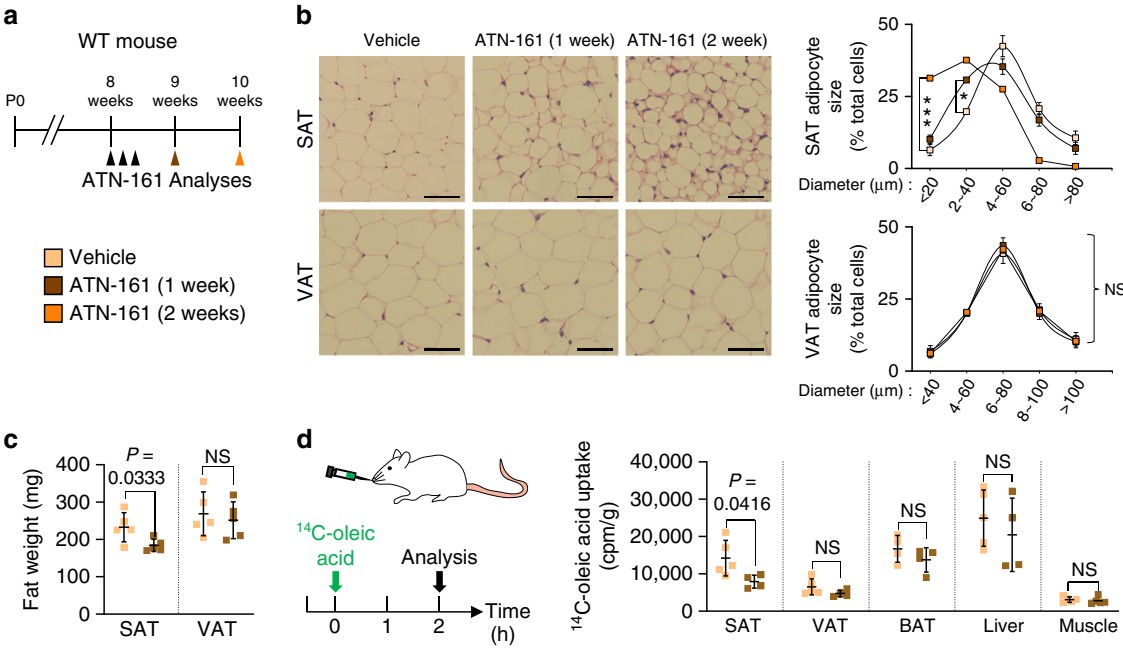

**Fig. 7 Integrin α5β1 drives subcutaneous fat distribution. a** Diagram depicting experimental scheme for integrin α5β1 antagonist treatment. 8-week-old WT mice were treated with integrin α5β1 blocking peptide (ATN-161, 30 mg/kg) and analyzed 1 or 2 weeks later. Black arrowheads indicate injections of ATN-161. **b** Representative H&E-stained images and comparisons in adipocyte size (diameter; μm) in SAT and VAT of ATN-161 treated mice. Four different mice of each genotype were randomly selected to determine the adipocyte size and data are presented as % of total cells. Horizontal bars indicate mean ± SD and *P < 0.05; ***P < 0.001 versus WT by two-tailed Student's t test. Scale bars, 30 μm. **c** Comparison of fat weight (mg) of SAT and VAT in ATN-161 treated mice. n = 5 for each group. Horizontal bars indicate mean ± SD and P values versus WT by two-tailed Student's t test. NS, not significant. **d** Diagram of experimental scheme and comparisons of [14]C-oleic acid uptake in different adipose tissues and metabolic organs between vehicle and ATN-161 treated mice. 8-week-old WT mice were dosed with ATN-161, and administered with [14]C-oleic acid 1 week after. n = 4 (WT BAT; ATN-161 SAT, VAT, BAT, and liver), 5 (WT SAT, VAT, liver, and muscle; ATN-161 muscle). Horizontal bars indicate mean ± SD and P values versus vehicle by two-tailed Student's t test. NS not significant.

process triggers fat accumulation in other fat depots and major glucose-consuming organs, leading to systemic insulin resistance with HFD, reminiscent of the pattern in diabetic-obese patients (Fig. 10).

Angpt2 is stored in repository granules of ECs called Weibel–Palade bodies, and rapidly released upon stimulation as an angiocrine factor[44,45]. Thus, it is intriguing that only adipocyte-specific but not EC-specific Angpt2 KO mice exhibited phenotypes in SAT. How does Angpt2 in adipocytes exert different effects from Angpt2 in ECs? Adipocytes do not possess Weibel–Palade bodies; thus, it is possible that the signals triggering Angpt2 release by adipocytes are distinct from those in ECs. For example, ECs release Angpt2 in inflammatory conditions to increase vascular permeability for immune cell infiltration[34,45]. On the other hand, we found a gradual increase in Angpt2 expression in adipocytes after a short-term high-fat regimen, which did not alter circulating Angpt2 levels. Thus, fat intake-induced Angpt2 released from adipocytes can stimulate FA uptake by neighboring ECs without systemic impact. In eliciting this outcome, integrin receptors face the basolateral side to bind with the extracellular matrix[46]. Although ITGα5β1 is on the basolateral side and CD36 resides on the luminal side, these seem to form a complex through transmembrane lipid rafts[39,40]. Thus, Angpt2 produced from the adipocytes may aggregate more easily than Angpt2 released from ECs circulating inside the lumen.

Another interesting feature of Angpt2 is its fat depot-specific effect. We found that adipocyte-specific deletion of Angpt2 affects SAT but not VAT. This selectivity can be explained by the distribution of its receptors; ECs in SAT highly express ITGα5β1 but barely express Tie2, whereas ECs in VAT have the opposite expression. EC-specific integrin KO mice phenocopied adipocyte-

specific Angpt2 KO mice further confirming that the integrin but not Tie2 is a key determinant of Angpt2 action on endothelial FA uptake in SAT. Therefore, endothelial heterogeneity among different fat depots mediates specific effects of Angpt2, which could be masked during systemic modulation of Angpt2 through pharmacological blockade or constant overexpression affecting other organs[23,24].

The differences in adipose depots' response to dietary fat intake could explain why VAT does not exhibit compensatory fat uptake. While SAT expands through hypertrophy of existing adipocytes, VAT responds through hyperplasia of newly generated adipocytes upon dietary fat intake[47,48]. This means that SAT expansion is mediated by fat intake itself whereas VAT expansion involves certain molecular cues to activate adipogenesis[49,50]. Meanwhile, it is well-known through clinical observation that subcutaneous obesity is more frequent in females and is less morbid than visceral obesity that is more frequent in males[2,51]. Whether adipocyte-derived Angpt2 and its role in endothelial FA transport is involved in the sexual dimorphism of body fat distribution needs to be studied. In order to do so, manipulating Angpt2 in a depot-specific manner could better define the role of Angpt2 in various adipose depots.

The important role of capillary ECs as a gatekeeper for fat trafficking has been demonstrated in various settings[8–11,17,52]. However, a study using imaging mass spectrometry to visualize cardiac FA uptake showed that the vascular wall is not a substantial barrier to the lipid movement into cardiomyocytes[53]. In addition, these authors reported that CD36 deficiency does not affect lipid entry into cardiomyocytes[53]. Yet another group recently reported that EC-specific deletion of CD36 leads to reduced lipid droplet accumulation in cardiomyocytes[16]. In

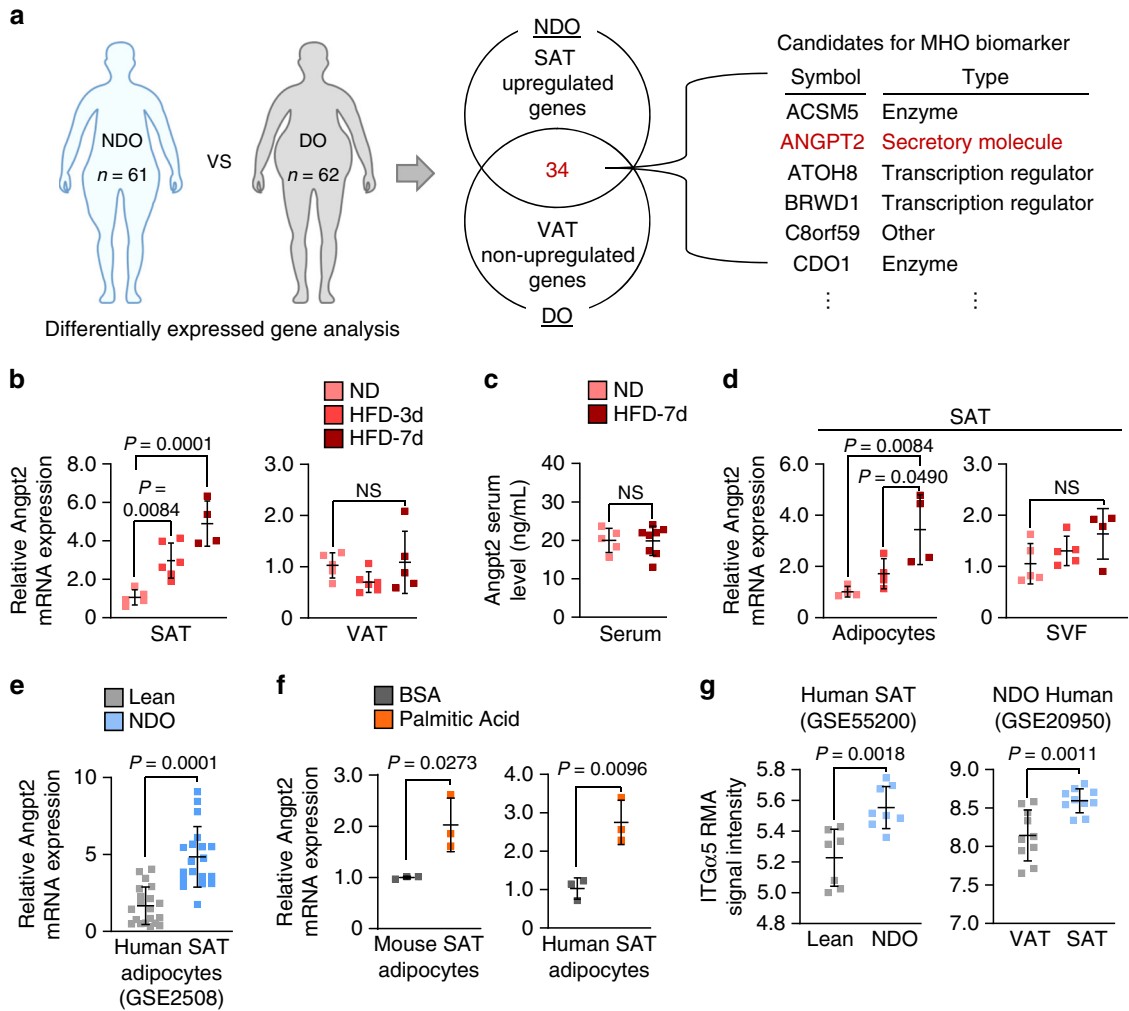

**Fig. 8 Angpt2 is preferentially expressed and upregulated in SAT upon lipid intake. a** Schematic diagram (left) for comparative transcriptomic analysis (Gene Expression Omnibus) between NDO ($n = 61$; nondiabetic obese) and DO ($n = 62$; diabetic obese) individuals. Venn diagram analysis (middle) indicating number of genes that are specifically upregulated in SAT (fold change > 1.5, $p < 0.05$) but not in VAT (fold change < 1.5) in NDO compared with DO individuals. List (right) of candidates among 34 genes that are specifically enriched in SAT of NDO individuals. Details of identified genes and analyzed gene sets are listed in Supplementary Tables 1, 2. **b** Comparisons of Angpt2 mRNA expression in SAT and VAT of WT mice fed with HFD for 3 days (HFD-3d) and 7 days. $n = 4$–6 mice/group pooled from two independent experiments. **c** Comparison of Angpt2 protein in serum of WT mice fed with HFD for 7 days. $n = 5$ (ND) or 8 (HFD-7d) mice/group pooled from two independent experiments. **d** Comparisons of Angpt2 mRNA expression in fractionated adipocytes and SVF in SAT of WT mice fed with HFD for 3 and 7 days. $n = 4$ (all Ad and SVF HFD-7d) or 5 (SVF ND and HFD-3d) mice/group pooled from two independent experiments. **e** Comparison of Angpt2 mRNA expression in adipocytes of SAT in lean ($n = 20$) and NDO ($n = 19$) human (GEO accession number: GSE2508; nondiabetic obese). **f** Comparisons of Angpt2 mRNA expression level in primary cultured adipocytes isolated from SAT of WT mice and human treated with either BSA alone or BSA-conjugated palmitic acid (400 μM) for 24 h under reduced serum (2%). Each dot indicates a mean of triplicate values from three independent experiments. **g** Comparison of ITGα5 mRNA expression in NDO and lean individuals (GEO accession number: GSE55200) and in SAT and VAT in NDO individuals (GEO accession number: GSE20950; nondiabetic obese). Each group, $n = 7$ (lean) or 8 (NDO); $n = 10$ (right panel). Unless otherwise denoted, horizontal bars indicate mean ± SD and $P$ values versus control by one-way ANOVA followed by Tukey's multiple comparison test (**b**, **d**) or two-tailed Student's $t$ test (**c**, **e**–**g**). NS not significant.

agreement, we found that CD36 is necessary for Angpt2-mediated endothelial FA uptake. Thus, it is possible that CD36 regulates FA uptake in a context-dependent manner. On the other hand, genetic mouse models for FATPs and their phenotypes regarding endothelial fat metabolism have not yet been reported. Thus, studies using animal models with genetic modifications in different FATPs in ECs will be useful to demonstrate their importance in regulating endothelial FA transport.

Through comparative transcriptomics on samples from NDO versus DO individuals, we identified Angpt2 as a potential adipokine that sustains metabolic health via regulation of body fat distribution. Accordingly, our adipocyte-specific Angpt2 KO mice demonstrated significant ectopic fat accumulation in the

BAT and in liver and skeletal muscle after HFD. This accumulation is accompanied by markedly reduced FA uptake and SAT size, indicating that SAT adipocyte-released Angpt2 is critical for proper fat distribution to prevent fat spillover to other organs. The question then arises: Can Angpt2–ITGα5β1 treatment be a therapeutic strategy to normalize fat distribution and treat obesity-induced metabolic disorders? The doses and administration route may be critical given that a constant or systemic increase in Angpt2 elicits other impacts, such as activation of angiogenesis and increased vascular permeability[23,54,55]. Alternatively, targeting the SAT endothelium would be a more attractive approach. Further investigation is required to test these possibilities genetically or

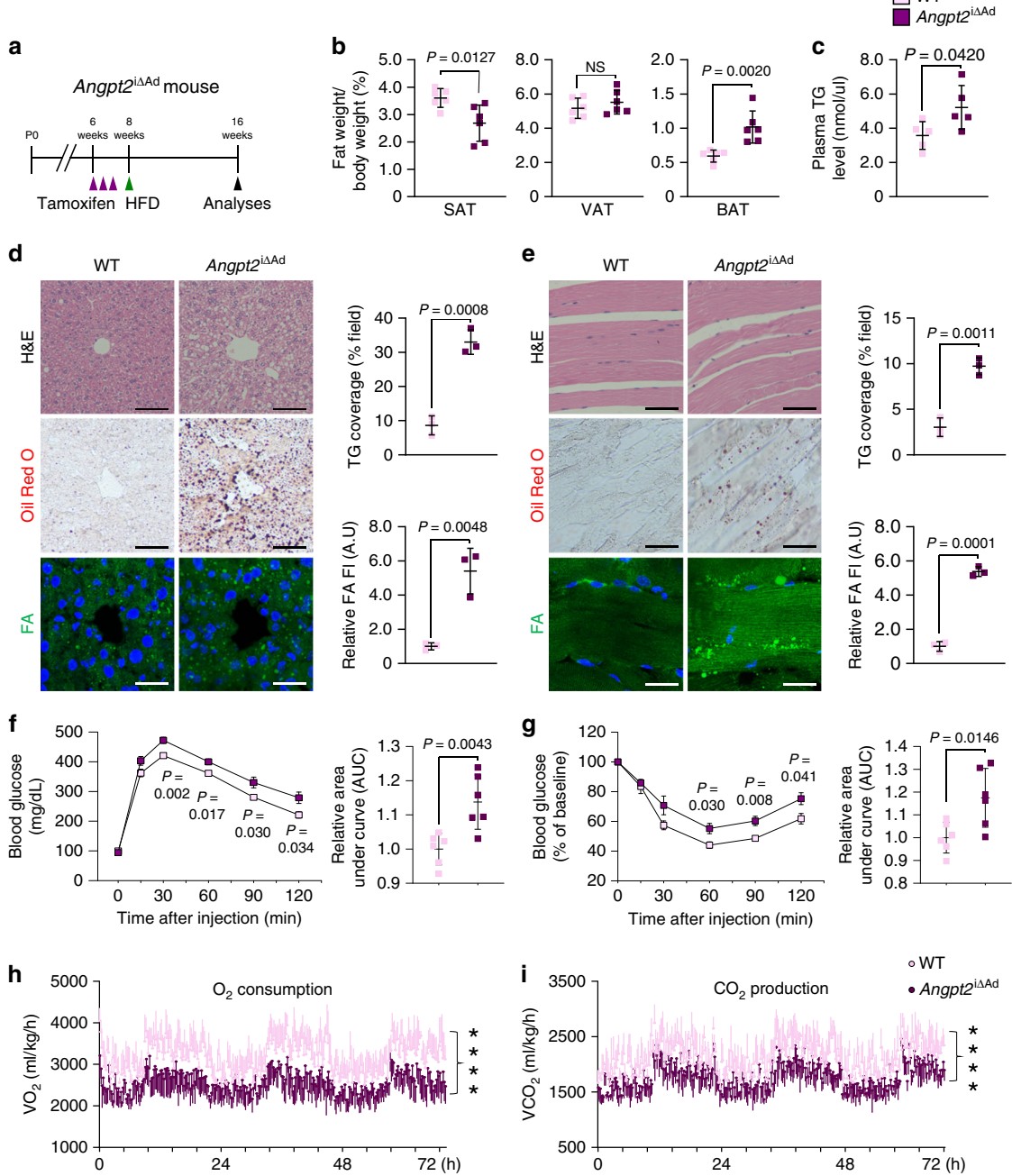

**Fig. 9 Adipocyte-derived Angpt2 prevents ectopic lipid-induced insulin resistance. a** Generation of *Angpt2*$^{iΔAd}$ mouse, inducible and specific deletion of *Angpt2* in adipocytes in 6-week-old mice, and analysis after 8 weeks of HFD. Purple arrowheads indicate daily injections of tamoxifen. **b** Comparisons of fat-to-body weight ratio in different adipose tissues between WT and *Angpt2*$^{iΔAd}$ mice fed with HFD. Each dot indicates a value obtained from one mouse and $n = 6$ mice/group pooled from two independent experiments. **c** Comparison of plasma triglyceride (TG) levels between WT and *Angpt2*$^{iΔAd}$ mice. Each dot indicates a value obtained from one mouse and $n = 5$ mice/group pooled from two independent experiments. **d** Representative images and comparisons of H&E-stained sections, Oil Red O staining and coverage (% of field), and BODIPY-FA staining and uptake (fluorescent unit, FI) of liver between WT and *Angpt2*$^{iΔAd}$ mice fed with HFD. Each dot indicates a mean of triplicate values from three independent experiments. Scale bars, 50 μm. **e** Representative images and comparisons of H&E-stained sections, Oil Red O staining and coverage (% of field), and BODIPY-FA staining and uptake (FI) of skeletal muscle between WT and *Angpt2*$^{iΔAd}$ mice fed with HFD. Each dot indicates a mean of triplicate values from three independent experiments. Scale bars, 50 μm. **f, g** Comparisons of intraperitoneal glucose tolerance test (IPGTT) and intraperitoneal insulin tolerance test (IPITT) with corresponding area under curve (AUC) between WT and *Angpt2*$^{iΔAd}$ mice fed with HFD. Each dot indicates a value obtained from one mouse and $n = 6$ mice/group pooled from three independent experiments. Horizontal bars indicate mean ± SEM. *$P < 0.05$; ***$P < 0.001$ versus WT by two-tailed Student's $t$ test. **h, i** Comparisons of $O_2$ consumption and $CO_2$ production. HFD fed WT and *Angpt2*$^{ΔAd}$ mice were subjected to metabolic cage assessment with CLAMS analysis. Each group, $n = 6$. Horizontal bars indicate mean ± SEM. ****$P < 0.0001$ versus WT by two-tailed Student's $t$ test. Unless otherwise denoted, horizontal bars indicate mean ± SD and $P$ values versus WT by two-tailed Student's $t$ test (**c**, **e**, **f**, **g**). NS not significant.

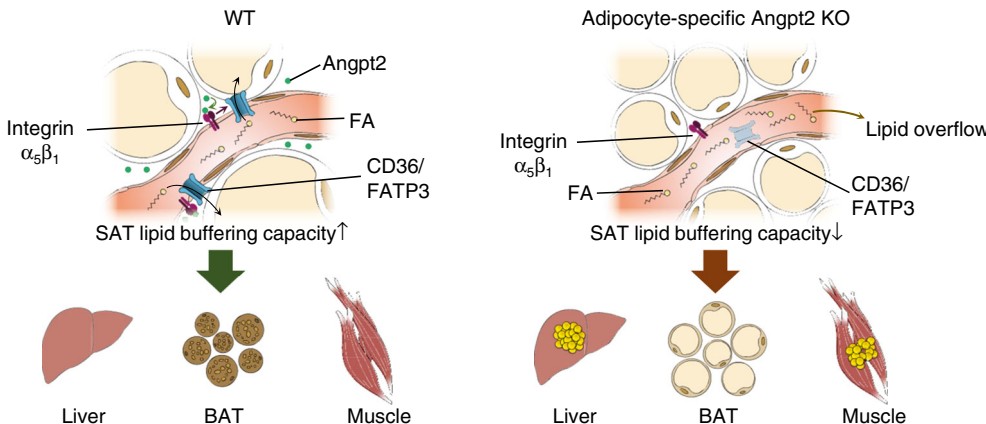

**Fig. 10 Endothelial-to-adipocyte fatty acid transport determines metabolic health.** Schematic diagram depicting Angpt2 produced from adipocytes could regulate endothelial FA transport via CD36 and FATP3 through integrin α5β1 signaling to accumulate FAs toward SAT. This process prevents ectopic fat accumulation into endocrine organs, and thus prevents insulin resistance.

pharmaceutically to open new therapeutic paths for metabolically healthy obesity.

## Methods

**Animals**. Specific pathogen-free C57BL/6J, *Adiponectin*-Cre, *Integrinβ1* flox/flox, *Tie2*-GFP, and RiboTag mice were purchased from Jackson Laboratory (Jackson Labs, Bar Harbor, ME). *Angpt2*-eGFP (Tg [Angpt2-EGFP] DJ90Gsat/Mmcd) were purchased from the Mutant Mouse Regional Resource Centers, *Angpt2* flox/flox [29], *Angpt2*-lacZ[56], *Adiponectin*-Cre-ER[T2] [57], and *VE-Cadherin*-Cre-ER[T2] [58] mice were transferred and bred in our specific pathogen-free animal facilities in Korea Advanced Institute of Science and Technology (KAIST). Mice were housed under 12 light/12 dark cycle, temperatures of 22 ± 2 °C with 50 ± 10% humidity. For all experiments, male mice aged 8-week-old under standard chow diet or 16-week-old under long term high-fat diet (HFD) were used. In order to induce Cre activity in the Cre-ER[T2] mice, 2 mg of tamoxifen was injected i.p. for 2 or 3 consecutive days from the indicated time points. For integrin α5β1-specific inhibition, mice were treated with ATN-161 (30 mg/kg)[35,59] for indicated time points and analyzed at indicated days. All mice were bred in our specific-pathogen-free animal facility and were fed normal chow diet (PMI LabDiet, St. Louis, MO) or HFD (60 kCal% fat, Research Diets, New Brunswick, NJ) with ad libitum access to water. Animal care and experimental procedures were performed under the approval from the Institutional Animal Care and Use Committee (IACUC; No. KA2013-39) of KAIST.

**Histological analyses**. Mice were anesthetized with a combination of ketamine (80 mg/kg) and xylazine (12 mg/kg) by intramuscular injection. All adipose tissues were from male adult mice. For sampling of tissues, inguinal white adipose tissue was used for SAT, epididymal white adipose tissue was used for VAT, interscapular BAT, and quadriceps (skeletal muscle) were used for this study. For hematoxylin and eosin (H&E) staining, indicated organs were fixed overnight in 10% paraformaldehyde (PFA) at 4 °C. After tissue processing using standard procedures, samples were embedded in paraffin and cut into sections followed by H&E staining. For immunofluorescence studies, harvested tissues were fixed overnight with 1% PFA in PBS at 4 °C, and whole mounted. After blocking with 5% goat or donkey serum (Jackson ImmunoResearch) in PBST (0.3% Triton X-100 in PBS for whole-mount method, 0.03% Triton X-100 in PBS for paraffin-sections) for 1 h at RT, whole-mounted or sectioned tissue was incubated overnight at 4 °C with the following primary antibodies (diluted at a ratio of 1:200 in blocking solution): anti-Perilipin (guinea pig polyclonal, 20R-PP004, Fitzgerald), anti-mouse CD31 (hamster monoclonal, 2H8, Millipore), anti-GFP (rabbit polyclonal, AB3080, Millipore), anti-cleaved caspase-3 (rabbit polyclonal, 9661, Cell Signaling), anti-UCP1 (rabit polyclonal, ab23841, Abcam), anti-Integrin α5β1 (rat monoclonal, BMB5, Millipore), anti- active Integrin β1 (rat monoclonal, 9EG7, BD Biosciences), anti- Integrin β1 (mouse monoclonal, 12G10, Abcam), anti-HA (rabbit polyclonal, H6908, Sigma-Aldrich), and anti-human CD31 (rabbit polyclonal, ab28364, Abcam). After several washes with PBST, the samples were incubated with the following secondary antibodies diluted at a ratio of 1:1000 in blocking solution (all from Jackson ImmunoResearch) for 2 h at RT: Cy3-conjugated anti-guinea pig antibody, Cy5-conjugated anti-hamster antibody, FITC-conjugated anti-rabbit antibody, FITC-conjugated anti-rat antibody, and FITC-conjugated goat antibody. Hoechst 33342 (Sigma-Aldrich) was used to detect nucleus, and borondipyrromethene (BODIPY, Invitrogen) was used to detect lipid accumulation.

To evaluate β-galactosidase activity, the tissues were incubated with a staining solution [5 mM potassium ferricyanide, 2 mM magnesium chloride, 5 mM potassium ferrocyanide, and 1 mg/ml 4-chloro-5-bromo-3-indolyl-β-D-galactopyranoside (X-gal) in PBS] at 37 °C for 16 h. For Oil Red O staining, slides were washed with PBS and 60% isopropanol, and incubated with filtered Oil Red O working solution for 50 min at room temperature. After staining, several washes with PBS and 60% isopropanol were performed to reduce nonspecific staining.

**Morphometric analysis**. Confocal microscopes (LSM 800 and LSM 880, Carl Zeiss) and stereomicroscope (Axiozoom V16) equipped with argon and helium–neon lasers were used to visualize fluorescence images. Morphometric analyses were performed with Image J software (NIH) or Zeiss image software (ZEN 2012). Histological sections of adipose tissues were stained with H&E and studied under 40-fold magnification to compare adipocyte size. Four to six different mice of each genotype were randomly selected to determine the adipocyte size by cross-sectioned area using Image J software. Cross-sectioned area were narrowed down into ranges of adipocyte size, and data were presented as percentage of adipocytes of each range of adipocyte size.

Vascular density was measured as CD31[+] vessel area divided by total measured area and presented as percentage.

**RNA extraction**. Total RNA was extracted from the samples using TRIzol® Reagent (Invitrogen) according to the manufacturer's instructions. For adipocyte and SVF isolation, adipose tissues were incubated in Hanks balanced salt solution (HBSS; Sigma-Aldrich) containing 0.2% collagenase type 2 (Worthington) for 30 min at 37 °C with constant shaking. After inactivating collagenase activity with 10% fetal bovine serum (FBS) containing Dulbecco modified eagle medium (DMEM), the cell suspension was filtered through a 40 µm nylon mesh (BD Biosciences), followed by centrifugation at 420 g for 5 min. Floating adipocytes were used as adipocyte fraction, remaining SVF pellet were isolated and further analyzed. 500–2000 ng of the RNA were reverse transcribed into cDNA using GoScript[TM] cDNA synthesis system (Promega, Madison, Wisconsin).

**Quantitative RT-PCR analyses**. Quantitative real-time PCR was performed using FastStart SYBR Green Master mix (Roche) and QuantStudio3 (Applied Biosystems) with the indicated primers. The real-time PCR data were analyzed with QuantStudio Software (Applied Biosystems). Results were calculated using the delta delta CT method[60], with *36b4* used for normalization of in vivo samples and *Gapdh* for normalization of in vitro samples. Primers for the quantitative real-time PCR are shown in Supplementary Table 3.

**Oxygen consumption rate**. The oxygen consumption rate was measured using the Seahorse XFe96 analyzer (Seahorse Bioscience) following the manufacturer's instructions. Briefly, ECAR and OCR were measured after primary cultured preadipocytes were stimulated with adipogenic cocktail on XFe96 microplates. Cells were maintained in non-buffered assay medium in a non-$CO_2$ incubator for 1 h before the assay. The Mito stress test kit (Seahorse Bioscience) was used to test the OCR under basal conditions in the presence of oligomycin (1.5 µM), the mitochondrial uncoupler carbonyl cyanide-4-(trifluoromethoxy)phenylhydrazone (FCCP; 1 µM), and the respiratory chain inhibitors rotenone and antimycin A (0.5 µM).

**Fatty acid uptake in vivo**. For measuring FA absorption in vivo using radioisotopes, mice were given a bolus dose of 2 mCi of $^{14}C$-oleic acid (Perkin Elmer) dissolved in 200 µl olive oil by oral gavage. After 2 h, indicated organs were dissected. The organs were incubated overnight at 50° in 1 ml tissue solubilizer (Solvable, Perkin Elmer), decolorized with 0.3 ml 30% hydrogen peroxide for 1 h in room temperature. We added scintillation solution (Ultima Gold, Perkin Elmer) to

each vial. Total radioactivity was measured by liquid scintillation using a Tri-Carb 2910 TR Liquid Scintillator (Perkin Elmer). $^{14}$C-oleic acid content was normalized to g of tissue.

For measuring FA absorption in vivo using fluorescent dyes, we revised a previously described method[20]. In brief, the appropriate amount of BODIPY fluorescent-conjugated FAs (BODIPY FL C16 and BODIPY 558/568 C12, Invitrogen) was calculated (0.5 mg/kg) for each mice and dissolved in control solution (HBSS supplemented with 20 mM HEPES and 0.2% FFA free BSA) in a total volume of 200 μl. Injection was performed intravenously, and tissues were harvested and snap frozen at indicated time points. Tissues were then homogenized in RIPA buffer, centrifuged and supernatant was used. Fluorescence intensity was measured ($\lambda$ex = 485/$\lambda$em = 520 nm) in black, 96-well flat-bottom plates using a fluorescence microplate reader (Tecan). We subtracted the fluorescence signal of each tissue from mice treated with control solution, and normalized by the weight of the extracted tissue. For both methods, those with insufficient FA intake in plasma were ruled out for data analysis.

**Cell culture**. Human Umbilical Vein Endothelial Cells (HUVECs) were cultured according to the manufacturer's protocols (Lonza, Walkersville, Maryland). In brief, the cells were cultured in a humidified atmosphere with 5% CO$_2$ at 37 °C in endothelial growth medium (EGM, Lonza) on cell culture plates without gelatin or fibronectin coating to minimize effects of integrin activation from other growth factors or matrix. The cells used were between passages 3–6.

For SVF induction, dissected SAT were chopped with scissors and incubated in digestion medium containing 0.2% collagenase type 2 (Worthington) for 30 min at 37 °C with constant shaking. The resulting suspension were lysed with ACK Lysing Buffer (Life Technologies) for 5 min at 37 °C, and centrifuged at 470 × $g$ for 5 min. After several washes, SVF pellet was resuspended in culture medium. Two days after confluent state, SVF cells were cultured with adipogenic differentiation medium. After 3 days, adipogenic differentiation medium was changed to maintenance medium.

Human primary subcutaneous pre-adipocytes (PCS-210-010; ATCC) were cultured in fibroblast basal medium (PCS-201-030; ATCC) added with fibroblast growth kit-low serum (PCS-201-041; ATCC) and differentiated using adipocyte differentiation toolkit (PCS-500-050; ATCC). MS1 mouse pancreas ECs (CRL-2279; ATCC) were cultured in DMEM medium supplemented with 5% fetal calf serum (FCS) at 37 °C and 5% CO$_2$.

For primary culture of ECs from SAT and VAT, we employed and modified a previously published method for culturing primary ECs of murine adipose tissues[37]. Dissected SAT and VAT were chopped with scissors and incubated in digestion medium containing 0.2% collagenase type 2 (Worthington) for 30 min at 37 °C on a shaker. The resulting suspension was lysed with ACK Lysing Buffer (Life Technologies) for 5 min at 37 °C and centrifuged at 470 × $g$ for 5 min. After several washes, cell suspension was filtered through 100 μm nylon cell strainer and washed. To enrich the EC fraction, cells were incubated for 20 min with anti-CD31 Microbeads (Miltenyi) and selected using AutoMACS (Miltenyi) according to the manufacturer's instructions. In order to maximize EC survival and culture, cells were incubated with EGM2 containing 10% FBS and 400 ng/ml of EC growth supplement (ECGS; Sigma). Culture-expanded monolayer of ECs under single passage that were validated to be CD31+ were used for all experiments.

For SVF induction, dissected SAT were chopped with scissors and incubated in digestion medium containing 0.2% collagenase type 2 (Worthington) for 30 min at 37 °C with constant shaking. The resulting suspension were lysed with ACK Lysing Buffer (Life Technologies) for 5 min at 37 °C, and centrifuged at 470 × $g$ for 5 min. After several washes, SVF pellet was resuspended in culture medium. Two days after confluent state, SVF cells were cultured with adipogenic differentiation medium. After 3 days, adipogenic differentiation medium was changed to maintenance medium.

Human primary subcutaneous pre-adipocytes (PCS-210-010; ATCC) were cultured in fibroblast basal medium (PCS-201-030; ATCC) added with fibroblast growth kit-low serum (PCS-201-041; ATCC) and differentiated using adipocyte differentiation toolkit (PCS-500-050; ATCC). MS1 mouse pancreas ECs (CRL-2279; ATCC) were cultured in DMEM medium supplemented with 5% FCS at 37 °C and 5% CO$_2$.

**Cell adhesion assay**. For cell adhesion assays, 48-well tissue culture plates were either uncoated or coated with 0.1% gelatin or 10 μg/mL fibronectin in PBS at 4 °C for 1 h, air dried, and rinsed once with PBS. After being serum deprived at 37 °C for 8 h, HUVECs were detached and washed twice, and plated onto the wells in serum-free medium containing 0.1% BSA at 5 × 10$^4$ per well. Cells in three wells of the quadruplicate were allowed to adhere to the coated/uncoated surface for 30 min, followed by four intensive washes to remove nonadherent cells, and incubated with 44 μM resazurin (#R7017, Sigma-Aldrich) in complete medium for 2 h. Resazurin fluorescence was then measured with a microplate reader (excitation 530 nm, emission 590 nm, cutoff 550 nm). Values were normalized to control.

**siRNA transfection**. For siRNA experiments, HUVECs were transfected with siR-NAs targeting human CD36 (5′-AAGAGGAACTATATTG-TGCCTCCTGT CTC-3′), FATP3 (Santa Cruz Biotechnology, SASI_Hs01_00100092), FATP4 (Santa Cruz Biotechnology, SASI_Hs01_00047-530), ITGβ1 (5′-TGATAGAT

CCAATGGCTTA-3′), ITGα5 (Bioneer, 1075709), ITGαV (Bioneer, 1075799), ITGβ3 (Bioneer, 1075875), ITGβ5 (Bioneer, 1075906), TIE2 (5′-GGCUAGUAAGAU-CAAUGGUdTdT-3′), NFATc1 (5′-CCCGUUCACGUCAGUUUCUAC GUCU-3′)or a scrambled control (5′-UAGCGACUAAACACAUCAA-3′) were used. Transfections of siRNA into the HUVECs were performed using Lipofectamine® RNAiMAX (Invitrogen, Waltham, Massachusetts) according to the manufacturer's protocol. Briefly, 20 nM of siRNA was transfected with RNAiMAX and knockdown was assessed by RT-PCR. Experiments were conducted 48 h after siRNA transfection.

**Fatty acid uptake and transport in vitro**. For measuring FA absorption in vitro, we revised a previously described method[9]. In brief, confluent HUVECs were transferred from a 10 cm dish to a non-coated, 96-well, black, clear-bottom plate (Corning, 3603), with empty corner wells for no-cell controls. After overnight incubation, the cells were serum-starved for at least 6 h. The cells were then treated with Mn$^{2+}$ to activate integrin motifs[21] with vehicle or human recombinant Angpt2 (2.5 μg/ml). Then, BODIPY-FA (Invitrogen, D3823), preincubated with fatty acid-free bovine serum albumin (BSA) (2:1 molar ratio) in PBS for 30 min in a 37 °C water bath, was added to the cells for 5–30 min at 37 °C. The BODIPY solution was washed out completely aspirated, and the cells were washed with 0.5% BSA in PBS for 1.5 min twice (100 μl per well). One percent PFA was added (100 μl per well) to minimize degradation, and intracellular fluorescence was measured (excitation 488 nm, emission 515 nm) immediately with a microplate reader (Tecan, BioTek). Readings from wells without BODIPY addition were used to subtract background signals. The cells were then incubated with HOECHST (100 μl per well) and was measured with a microplate reader (excitation 350 nm, emission 461 nm) to normalize BODIPY signals to cell number. BODIPY FL C16 (D3821) and BODIPY FL C5 (D3834) were purchased from Invitrogen. To block integrin α5β1, 10 μM ATN-161[35] were treated 15 min prior to Angpt2 addition. To activate integrin α5β1-specific motif, 10 μg/ml SNAKA-51[22] were treated 15 min prior to Angpt2 addition. For all experiments, a fresh batch of recombinant Angpt2[61] were treated in freshly isolated HUVECs (less than four passages).

For measuring FA transport in vitro, we revised a previously described method[8]. HUVECs were grown on a 24-well 0.4 μm trans-well inserts (SPL, 35024), and they were grown for 2–3 days until the cells formed compact layers. Phenol red-free ECBM (Promocell, C-22215) were used to minimize overlapping fluorescence of medium samples that were collected from the bottom chamber at the indicated time points. Fifty microliters of medium sample was measured using a fluorescence plate reader (excitation 488 nm, emission 515 nm).

**In vitro vascular permeability assay**. For assessment of endothelial layer permeability, we modified the manufacturer's instructions of in vitro vascular permeability assay (Millipore). Briefly, HUVECs were grown until confluence for 2–3 days on trans-well inserts. FITC-dextran solution (Millipore) was added to the upper chamber and transferred solution was measured.

**In vivo vascular permeability assay**. Vascular leakage was analyzed after i.v. injection of 100 μl of FITC-conjugated dextran (4 mg/ml, 70 kDa, Sigma-Aldrich) 5 min before sacrifice. Mice were anesthetized and perfused by intracardiac injection of 1% PFA to remove circulating dextran.

**mRNA isolation using RiboTag method**. RiboTag mouse was used to isolate polysome-bound mRNAs of EC with minor modification from previously described method[32]. Briefly, tissues were harvested and immediately snap frozen. Then, polysome buffer (50 mM Tris, pH 7.5, 100 mM KCl, 12 mM MgCl$_2$, 1% Nonidet P-40, 1 mM DTT, 200 U/ml RNasin, 1 mg/ml heparin, 100 μg/ml cyclohexamide, and 1× protease inhibitor mixture) were added to each sample and homogenized using Precellys lysis kit (Bertin). For IP against HA, anti-HA antibody-conjugated magnetic beads (MBL, M180-11) were added to the supernatant after centrifugation for 10 min at 13500 × $g$ 4 °C, and incubated on a rotating shaker at 4 °C overnight. Beads were washed for four times with high-salt buffer (50 mM Tris, pH 7.5, 300 mM KCl, 12 mM MgCl$_2$, 1% Nonidet P-40, 1 mM DTT, and 100 μg/ml cyclohexamide) and resuspended in 350 μl of RLT plus buffer with β-mercaptoethanol. Total RNAs were extracted using the RNA isolation mentioned in methods. The quality and quantity of the RNA samples were analyzed using Agilent 2100 Bioanalyzer with RNA 6000 pico kit (Agilent), and further performed to RNA-seq.

**RNA-seq analysis of RiboTag-captured mRNA**. For RNA-seq data analysis, four to five biological replicates of mRNA isolated by RiboTag method were used for analysis. The normalized count values were processed based on Quantile normalization method using the Genowiz™ version 4.0.5.6 (Ocimum Biosolutions) and used for heatmaps and bioinformatics analysis. RNA-Seq gene expression heatmap was generated with Multiple Experiment Viewer from The Institute of Genomic Research. The indicated expression ratio in a heatmap reflects the normalized count of each replicate apart from the mean gene expression value over all condition.

**Bioinformatics**. For RNA-seq experiment, SENSE 3′ mRNA-Seq Library Prep Kit (Lexogen, Inc.) were used according to the manufacturer's instructions. In brief, each 500 ng total RNA were prepared and an oligo-dT primer containing an Illumina-

compatible sequence at its 5′ end was hybridized to the RNA and reverse transcription was performed. After degradation of the RNA template, second strand synthesis was initiated by a random primer containing an Illumina-compatible linker sequence at its 5′ end. The double-stranded library was purified by using magnetic beads to remove all reaction components. The library was amplified to add the complete adapter sequences required for cluster generation. The finished library is purified from PCR components. High-throughput sequencing was performed as single-end 75 sequencing using NextSeq 500 (Illumina, Inc.). For RNA-Seq, SENSE 3′ mRNA-Seq reads were aligned using Bowtie2 version 2.1.0. Bowtie2 indices were either generated from genome assembly sequence or the representative transcript sequences for aligning to the genome and transcriptome. The alignment file was used for assembling transcripts, estimating their abundances and detecting differential expression of genes. Differentially expressed gene were determined based on counts from unique and multiple alignments using EdgeR within R version 3.2.2 (R development Core Team) using BIOCONDUCTOR version 3.0. The RT (Read Count) data were processed based on Quantile normalization method using the Genowiz™ version 4.0.5.6 (Ocimum Biosolutions).

**Immunoblotting**. For immunoblot analysis, cells were lysed on ice in RIPA lysis buffer supplemented with protease and phosphatase inhibitors (Roche). Cell lysates were centrifuged for 10 min at 4 °C, $16,000 \times g$. Protein concentrations of the supernatants were quantitated using the detergent-insensitive Pierce BCA protein assay kit (Thermo Scientific, 23227). Aliquots of each protein lysate (10–20 µg) were subjected to SDS polyacrylamide gel electrophoresis. After electrophoresis, proteins were transferred to nitrocellulose membranes and blocked for 30 min with 5% skim milk in TBST (0.1% Tween 20 in TBS). For phosphorylated protein detection, membranes were blocked with 2% BSA in TBS. Primary antibodies were incubated overnight at 4 °C. After washes, membranes were incubated with anti-rabbit (CST, 7074) or anti-mouse (CST, 7076) secondary peroxidase coupled antibody for 1 h at RT. Target proteins were detected using ECL western blot detection solution (Millipore, WBKLS0500). The uncropped and unprocessed scans with marker positions of all blots were included in the Source data file.

**Immunoprecipitation**. Cells were lysed in NETN lysis buffer (20 mM Tris-HCl pH 7.4, 100 mM NaCl, 1 mM EDTA, 0.5% Nonidet P-40) with protease inhibitors (Roche). Antibody was added to the cleared lysate and incubated overnight. Then, 30 µl of protein A/G agarose (Pierce) was added to the lysate, incubated for 2 h, washed with NETN buffer three times, and boiled in Laemmli's sample buffer. The samples were then subjected to SDS-PAGE gels for western blot analysis.

**In situ proximity ligation assay**. Cells cultured on confocal dishes were fixed with 4% PFA for 20 min at room temperature, permeabilized and incubated with primary antibodies at 4 °C. All primary antibodies were used at a 1:200 dilution. For in situ proximity ligation assay, protein–protein interactions between CD36 (Thermo) and Integrin β1 (abcam) were detected with secondary proximity probes (Anti-Rabbit Plus and Anti-Mouse Minus) according to the Duolink In Situ Fluorescence protocol (Sigma-Aldrich).

**Stable expression of FATP3 in HUVECs**. The cDNA sequence of human FATP3 was cloned into FUtdTW vector (addgene) containing td-Tomato sequence. Sequencing confirmed that no errors were introduced. Lentivirus production was performed with Lenti-X cells co-transfected with FUtdTW-FATP3, Delta 8.2 and Ampho plasmids using Lipofectamine LTX (Invitrogen). At day 3 after transfection, culture supernatants were collected and centrifuged at $450 \times g$ for 10 min to remove cell debris. Supernatants were filtered with 0.22 µm syringe filter and concentrated by using Centricon filters (30 kDa cutoff, Amicon). Lentiviral particles were transduced into HUVECs with hexadimethrine bromide for 20 h. Two days after lentiviral transduction, HUVECs were stably expressed, giving rise to a cell population with FATP3 expression. Cells were used immediately for experiments.

Live cell imaging was performed using the incubator chamber equipped with optimal environment settings of 5% $CO_2$ and 37 °C, live cell imaging was performed and recorded for 30 min by microscope (Cell Observer, Carl Zeiss).

**Sampling of human adipose tissues**. Human SATs were collected from female patients (ages 39–56) undergoing breast reconstruction after mastectomy for breast cancer. The Institutional Review Board of Seoul National University Hospital (1708-043-876) approved experimental procedures with human adipose tissue specimens. All human samples were collected in an unbiased manner by the tissue bank of Seoul National University Hospital, Seoul, Korea, with the informed consents from the donors following the bioethics and safety regulations.

**Comparison of nondiabetic obese and diabetic-obese patients**. For the analysis of differentially expressed genes in NDO versus DO, each gene expression data were collected from NCBI-GEO (GSE20950, GSE29226, GSE29231, GSE16415, GSE71416), a publically available database repository of high-throughput gene expression data. Collected datasets were annotated with official gene symbols, and normalized (log2). Differential expressed genes were analyzed separately in each gene sets. Among each datasets, common genes that were specifically upregulated

in SAT (fold change > 1.5, $p < 0.05$) were narrowed down with genes that were not upregulated in VAT (<1.5). Datasets including GEO reference number, sample number, organ, normalized Angpt2 expression, and status of patient (diabetic, obese) are indicated in Supplementary Table 2. Gene classification analysis was performed with Ingenuity Pathway Analysis (IPA, Qiagen) and indicated in Supplementary Table 1. For analysis of Angpt2 mRNA expression in human SAT, indicated gene sets was annotated and each value for Angpt2 expression was normalized (log2). Relative expression of Angpt2 in human SVF was compared between human adipocytes (GSE80654), and adipocytes from NDO individuals (BMI 55 ± 8) were compared with lean individuals (GSE2508). Relative expression of ITGα5 in NDO SAT was compared with lean individuals (GSE55200), and SAT from NDO individuals (obese, nondiabetic) were compared with VAT (GSE20950).

**Biochemical analysis of serum**. A 0.5 ml blood sample harvested in Vacutainer tubes (BD) were centrifuged for 20 min at $2000 \times g$ t 4 °C twice. The plasma activity triglyceride (TG) was measured using an automated analyzer (VetScan, Abaxis, CA, USA). To measure circulating Angpt2, mouse/rat Ang2 ELISA kit was used (R&D Systems) according to the manufacturer's instruction and measured using a Spectra MAX340 plate reader (Molecular Devices). To measure plasma level of FFA, mouse/rat/human FFA kit was used (abcam) according to the manufacturer's instruction using a Spectra MAX340 plate reader (Molecular Devices).

**Intraperitoneal glucose/insulin tolerance test**. For intraperitoneal glucose tolerance test, mice fasted for 16 h were injected with D-glucose (2 g/kg) (Sigma-Aldrich) intraperitoneally. For intraperitoneal insulin tolerance test, mice fasted for 4 h were injected with insulin (1 U/kg) (Sigma-Aldrich) intraperitoneally. For analysis of blood glucose concentrations, blood was collected from the tail vein at 0, 15, 30, 60, 90, and 120 min after insulin administration, and glucose was measured with a glucometer (Gluco Dr. Plus, All Medicus).

**Statistics and reproducibility**. Sample sizes were chosen on the basis of standard power calculations (with $\alpha = 0.05$ and power of 0.8) performed for similar experiments and statistical methods were not used to predetermine sample size. No samples were excluded from the analysis. Unless otherwise indicated, experiments were replicated at least once for all analyses and number of reproductions of each experimental finding is described in each figure legend. All attempts at experimental replication were successful. Animals from different cages, but within the same experimental group, were selected to assure randomization. Experiments involving in vitro and in vitro study was assured randomization through double-blind experiments. The investigators were not blinded during experiments involving long term HFD challenge due to clear appearance of body mass changes among the groups. However, two independent investigators have performed most of experiments in parallel and administration of chemicals was carried out as blinded experiments. Data are presented as mean ± standard deviation (SD) or mean ± standard error of the mean (SEM). Statistical differences between the means were compared by the two-tailed, unpaired $t$ test for two groups, or determined using one-way ANOVA followed by Tukey's multiple comparison test for multiple groups, unless otherwise noted. Statistical analysis was performed with PASW Statistics 18 (SPSS) or Prism 7 (GraphPad). Statistical significance was set to $P$ value < 0.05.

**Reporting summary**. Further information on research design is available in the Nature Research Reporting Summary linked to this article.

## Data availability

The RNA-seq data are available in the European Bioinformatics Institute (EMBL-EMI's) ArrayExpress under the accession number E-MTAB-6161. Other transcriptomic datasets analyzed in this study can be retrieved from the GEO repository under the accessions GSE20950, GSE29226, GSE29231, GSE16415, GSE71416 for the comparison of NDO vs. DO, GSE80654 for human SVF and adipocytes, GSE2508 for NDO vs. lean human adipocytes, GSE55200 for NDO vs. lean human SAT, and GSE20950 for NDO human SAT vs. VAT datasets. The source data underlying all Figs. and Supplementary Figs. are provided as a Source Data file. A reporting summary for this article is available as a Supplementary Information file. All other data that support the findings of this study are available from the corresponding author upon reasonable request.

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

## Acknowledgements

We thank Intae Park for proof reading of paper, Sujin Seo, Hyun Tae Kim, Jeomil Bae, and Taechang Yang for their technical assistances, Professor Yoshiaki Kubota (Keio University) for providing *VE-Cadherin*-Cre-ER[T2] mice, and Dr. Nicholas Gale (Regeneron

Pharmaceuticals) for providing *Angpt2-LacZ* reporter knock-in mice. H.B. is supported by NRF (National Research Foundation of Korea) Grant funded by the Korean Government (NRF-2014-Fostering Core Leaders of the Future Basic Science Program/Global Ph.D. Fellowship Program). This study was supported by the Institute for Basic Science funded by the Ministry of Science, ICT and Future Planning, Korea (IBS-R025-D1, G.Y.K.).

## Author contributions

H.B. designed, organized, and performed the experiments, generated the figures, and wrote the paper. K.Y.H. provided human samples and critical comments on this study. C.L., S.L., and K.C. provided important idea and technical supports. C.J. and H.J.L. supervised and participated in paper preparation. S.O. provided the mice and critical comments on this study. Y.H. and G.Y.K. designed, organized, supervised the project, and wrote the paper.

## Competing interests

The authors declare no competing interests.
