## [Peer Review File · Nature Communications]

Reviewers' comments:

Reviewer #1 (Remarks to the Author):

This study strives to demonstrate that subcutaneous adipose tissue is a major producer of Angpt2, and which exerts functional control over fatty acid uptake by endothelial cells via activation of $\alpha 5\beta 1$, leading to complex formation with CD36 and fatty acid uptake. This is an intriguing concept. However, there are several conceptual issues that make this difficult to understand. 1. The authors state that $\alpha 5\beta 1$ is preferentially on the abluminal surface of the endothelial cells. CD36 is on the luminal surface to facilitate fatty acid uptake from plasma. How then can a CD36 complex to integrin $\alpha 5\beta 1$ actually enhance fatty acid uptake from the plasma? 2. The authors have shown that these events are specific to SAT, yet how this specificity is achieved is not established. Why does SAT and not VAT or BAT produce Angpt2? Specific comments follow below:

Methodology issues:

1. Model characterization is inadequate - extent and specificity of deletion not demonstrated in any of the mouse models.
2. siRNA – inadequate details of experimental conditions (amount of siRNA, timing etc)
3. HUVEC plated without gelatin or fibronectin – how good is adherence?
4. N values – adipocyte size measured from only 3 mice per genotype – inadequate n value. Cell size data needs to be presented as an average per animal, rather than showing all individual counts. What are the area units? Why is the VAT adipocyte area less than 0? These numbers do not make sense – should be presented as adipocyte cross-sectional area. This is a major aspect of the study and thus data must be presented in a more rigorous and understandable way.
5. All experiments conducted on male mice – given the differential role of SAT in male and female, it would be valuable to compare sexes.
6. Normalization of RNA – relative expression was calculated by what procedure?
7. The timing of analysis of adipose tissue is different in every model (after 4 weeks for Angpt2 deletion, after 1 week for $\beta 1$ -deletion, after 1-2 weeks for $\beta 1$ blocking peptide). What is the rationale for these different time points?

Data issues:

1. Characterization of vasculature inadequate (Fig. 1 j) – what is being assessed? Relative intensity is not explained – how does this relate to number of vessels/vessel density? It appears that capillary number is increased in the Ang2 deficient mice. Since vascularization is intimately related to transport capacities of various tissues, it is not possible to make conclusions in the absence of a more thorough characterization of capillarization.
2. Fig 2b Why does VAT not experience fatty acid uptake similar to BAT? This is not explained by the data presented. The authors should discuss possible reasons, as the bystander effect does not seem to apply here.
3. The data indicate low mRNA expression of Tie2 in SAT (Fig. 3) appear to be at odds with other studies. And would have major implications for any studies using Tie2-Cre deleter models to investigate SAT. Further documentation of the expression of Tie2 in the SAT is warranted. Furthermore, what is the level of Tie2 expression on the HUVEC used in the experiments?
4. The Ribotag strategy is innovative. Analysis of these samples for RNAseq hardly justified since only 3 target genes are shown. Methodology does not adequately explain how the RNAseq data are displayed...what have they been normalized to?
5. Does Angpt2 deletion affect integrin expression in EC, or vice versa? Does HF diet impact integrin $\alpha 5\beta 1$ expression?
6. In vitro experiments would have been strengthened by using a microvascular EC from adipose tissue (since vascular bed-specific endothelial cell phenotype is well established) instead of HUVEC.

7. Regarding the in vitro data showing CD36, integrin $\alpha 5\beta 1$ interaction (Fig.5): Is there evidence that these molecules remain on the cell surface? They appear to be internalized into vesicles. What is the duration of time of this association following Angpt2 stimulation? Is CD36 on the cell surface initially? Why does siCD36 not affect basal FFA uptake?
8. A previous paper (An et al. eLife 2017) shows decreased angpt2 expression with HFD on SAT whereas this study shows that HFD increases angpt2 with short (3-7 days) and longer (3 mo) periods. The authors should discuss this discrepancy. Furthermore, An et al. (2017) have already shown that inactivation of Angpt2 exacerbates HFD induced metabolic defects and impairs lipid disposal, which was associated with TG accumulation in the liver.
9. Data for Human geoset analyses is poorly described, making it difficult to draw conclusions. What is the relevance of values shown in Fig.8g.
10. Previous studies using endothelial-specific depletion of beta1 integrin have noted severe permeability/vascular integrity issues. An in vivo assessment of vascular permeability is necessary to document the extent to which permeability might be altered in these mice.

Writing issues:

1. Many parts of the results are inadequately described within the text.
2. The discussion is inadequate in depth and scope.

Reviewer #2 (Remarks to the Author):

This very interesting manuscript shows for the first time that angiopoietin-2 controls uptake of fatty acids into subcutaneous but not visceral adipose tissue. This is achieved by signaling through integrins on endothelial cells which subsequently promote proper localization of fatty acid transport proteins, thereby facilitating endothelial transport of fatty acids from blood plasma to adipocytes. The authors show that there is differential Angpt2 expression in obese vs. lean subjects and that this may contribute to insulin resistance.

The manuscript is very well written, succinct and clear. The Material and Methods are well described. The figures are of high quality.

If revised properly, this work will significantly advance the field. It demonstrates a key example of organ-specific angiocrine signaling that controls metabolism.

Major points:

- 1) The authors demonstrate nicely increased Angpt2 in a high fat diet model. Is there also a physiological relevance of this study? One should determine Angpt2 and its receptor levels in adipose tissue upon feeding/fasting cycles and whether this affects expression of CD36 and FATP3.
- 2) Fig 2A and Fig 6C: how is this fatty acid uptake assay controlled? Do the authors measure the initial uptake into the blood stream that should be equal in both groups? How is the FA uptake into liver?
- 3) Is fatty acid production by the liver affected by Angpt2? In addition, what are the levels of triglycerides and free fatty acid in plasma under normal chow conditions?
- 4) Fatty acid uptake is also under control of insulin. What are insulin and glucose levels under basal conditions? Are there any changes in pancreas morphology.
- 5) A striking feature of this study is the clear organ specificity of Angpt2 action what the authors can convincingly correlate with the differential mRNA expression patterns of the integrin receptors. It

would be helpful to show the differential protein expression also. Therefore integrin staining should not only be shown for SAT (fig. 3G) but also VAT and BAT.

6) Another striking aspect of this study is that only Angpt2 derived from adipocytes but not from endothelial cells is important for FA transport? The authors speculate that endothelial release of Angpt2 occurs into the blood stream whereas adipocytes release Angpt2 to the basolateral membrane of endothelial cells. This is intriguing. It would be very interesting to see where the integrin receptors are located on endothelial cells (apical or basolateral) to substantiate this.

7) The changes in BAT are not convincingly explained. Are there any indications for altered thermogenesis?

8) The authors show immunohistochemistry of UCP1 in WAT. Is this sufficient to rule out quantitative changes? Western blot might be more appropriate. Also O₂ consumption of isolated WAT would be helpful for better interpretation.

9) Some of the observed changes could also be achieved due to altered immune cell infiltration. As Angpt2 is involved in this process, the authors should determine myeloid and lymphocyte infiltration rates in the various adipose tissue depots.

10) Fig 6 C: quantification of microvessel density is missing.

Reviewer #3 (Remarks to the Author):

The paper by Bae et al. provides substantial experimental evidence that the angiopoietin-2–integrin $\alpha 5\beta 1$ pathway is distinctively involved in a complex crosstalk between adipocytes and endothelial cells of subcutaneous fat, allowing delivery and storage of circulating fatty acids to subcutaneous adipocytes. Inadequate lipid storing abilities of subcutaneous fat is now regarded as a major pathophysiological mechanism leading over time in obese patients to high circulating lipid levels, abnormal lipid storage in target organs, cell lipotoxicity and onset of several metabolic complications of obesity, in particular insulin resistance and diabetes. Results are novel and, in view of the current wide medical interest for obesity and related diseases, the present findings will be of high interest for a wide audience of scholars. In my opinion, the experimental approaches involving both in vitro studies and animal models are well conceived and the methods are adequate and sufficiently detailed to allow a researcher to reproduce the experiments. Statistical analyses are appropriate. I only have one major concern and some minor suggestions to improve the paper.

The phenotypical characterization of *angpt2iΔAd* mice challenged with high-fat diet is a bit scant. Given the “whitened” morphological aspect of interscapular brown fat, the authors should measure the expression of UCP1 and other thermogenic markers in this depot and compare the results obtained with the respective controls. In addition, some obesity-linked inflammatory markers should be measured in the visceral depots to substantiate this model of obesity. Finally, evaluation of at least leptin and adiponectin blood levels in these mice could offer additional information to the dysmetabolic condition and insulin resistance present in these mice.

Minor points:

- Please, specify what you mean for visceral adipose tissue in mice. Is it the epididymal fat depot?

- The brown adipose tissue depot you analyzed is the interscapular (not "intrascapular").
- Even whether brown fat expresses much less angiotensin-2 than subcutaneous white adipose tissue it should be clearly stated (and discussed) that the use of adiponectin-Cre mice involves knocking out angiotensin-2 not only in white adipocytes but also in brown adipocytes.
- In Fig. 3f-h is not clear what is obtained by immunohistochemistry and what is obtained by using eGFP expressing mice.
- In Supplementary Figure 3 b liver pics are not comparable (one shows hepatocytes around a portal space, the other shows hepatocytes around a centrolobular vein)
- Please, explain in a few words what the vascular permeability test is.
- Data from humans on the expression of angiotensin-2 in non-diabetic obese patients are interesting; however, a reader also wonders what happens in diabetic obese patients. Are some data available?
- Please, specify in simple words what Weibel–Palade bodies are.
- I would expect the authors to discuss their results also in view of the well-known clinical observation that pear-shaped obesity that is more frequent in females and linked to enlargement of some subcutaneous depots is less morbid than the apple-shaped obesity that is more frequent in males and due to fat accumulation in visceral depots.

Antonio Giordano

Detailed Point-by-Point Response to Reviewers' Comments

We deeply appreciate the editor's and reviewers' thoughtful, critical, and constructive comments, which have undoubtedly provided us with valuable opportunities to improve our work. We have performed additional experiments and revised the manuscript to address the issues raised by the reviewers. To readily keep track of the changes that were made in revising our manuscript, we have highlighted them in blue except for the changes in figure legends.

Reviewer #1:

(Remarks to the Author)

This study strives to demonstrate that subcutaneous adipose tissue is a major producer of Angpt2, and which exerts functional control over fatty acid uptake by endothelial cells via activation of $\alpha 5\beta 1$, leading to complex formation with CD36 and fatty acid uptake. This is an intriguing concept. However, there are several conceptual issues that make this difficult to understand. 1. The authors state that $\alpha 5\beta 1$ is preferentially on the abluminal surface of the endothelial cells. CD36 is on the luminal surface to facilitate fatty acid uptake from plasma. How then can a CD36 complex to integrin $\alpha 5\beta 1$ actually enhance fatty acid uptake from the plasma? 2. The authors have shown that these events are specific to SAT, yet how this specificity is achieved is not established. Why does SAT and not VAT or BAT produce Angpt2? Specific comments follow below:

Response: We appreciate these critical comments. To address these issues, we first provide evidence that the complex formation of ITG $\alpha 5\beta 1$ and CD36 is mediated by lipid rafts, which enables abluminal ITG $\alpha 5\beta 1$ to bind to luminal CD36 through transmembrane lipid rafts. We explained this important issue in more detail in the response to the comment 14 by Reviewer 1.

Second, we believe that the specificity of ITG $\alpha 5\beta 1$ -mediated FA uptake in SAT is associated with the organotypic characteristics of both adipocytes and endothelial cells. It is well known that adipocytes in SAT are genetically different to VAT and BAT (Gesta et al., *Cell*, 2007; Schoettl et al, *J Exp Biol.*, 2018). These characteristics enable them to respond differently to fat intake, as we addressed in comment 9 by Reviewer 1. While adipocytes of SAT expand through hypertrophy of existing adipocytes in response to fat intake, VAT respond through hyperplasia (adipogenesis), and BAT burns excessive fat. In relation to this knowledge, we speculate that adipocytes in SAT sense excessive fat intake through NFATc1 (Shah and Brownlee, *Circ Res.*, 2016), which is known to regulate Angpt2 (Minami et al., *Cell Rep.*, 2013). Accordingly, we examined a similar pattern of SAT-specific increase in NFATc1 activity by high fat intake (Reviewer only Fig 1). In case of endothelial cells, preferential expression of ITG $\alpha 5\beta 1$ in SAT enables this organotypic behavior. We addressed about this issue in more detail in comment 5 for Reviewer 2.

Reviewer only Fig. 1

Comparisons of *Nfatc1* mRNA expression in fractionated adipocytes and SVF in SAT and VAT of WT mice fed with HFD for 3 days and 7 days. Each dot indicates a value obtained from one mouse; n = 3~5 mice/group pooled from two independent experiments. Horizontal bars indicate mean \pm SD and ***p < 0.001 vs. ND; ###p < 0.001 vs. ND by one-way ANOVA followed by Tukey's multiple comparison test. NS, not significant.

Methodology issues:

Comment 1: Model characterization is inadequate - -extent and specificity of deletion not demonstrated in any of the mouse models.

Response: In order to address the extent and specificity of gene deletion in the mouse models used in this study, we provide the following evidence. First, in adipocyte-specific deletion of *Angpt2*, we examined 88.7% decrease in *Angpt2* in the adipocyte fraction (Fig. 1b), while endothelial-specific deletion resulted in 64.9% decrease in *Angpt2* in stromal vascular fraction (Fig. 1m). Moreover, we confirmed the deletion of *ITG β 1* in EC in *ITG β 1 Δ EC* mice (Supplementary Fig. 9a) as well as the decreased expression of *ITG α 5 β 1* by ATN-161 (integrin α 5 β 1 antagonist) treatment (Supplementary Fig. 10a). We included these additional findings and their descriptions into the revised manuscript accordingly as below.

Fig. 1

b Comparisons of *Angpt2* mRNA expression in fractionized adipocytes (Ad) of SAT in WT and *Angpt2^{iAd}* mice. Each dot indicates a value obtained from one mouse; n = 3 mice/group. Horizontal bars indicate mean \pm SD and P values versus WT by two-tailed Student's t-test.

m Comparisons of *Angpt2* mRNA expression in stromal vascular fraction (SVF) of SAT in WT and *Angpt2*^{iΔEC} mice. Each dot indicates a value obtained from one mouse; n = 3 mice/group. Horizontal bars indicate mean ± SD and P values versus WT by two-tailed Student's t-test.

Supplemental Fig 9.

a Representative images of active ITGβ1 expression in EC of SAT in WT and *ITGβ1*^{iΔEC} mice. Scale bars, 50 μm.

Supplemental Fig 10.

a Representative images of ITGα5β1 expression in EC of SAT in mice treated with vehicle or ATN-161 (30mg/kg) for 1wk. Scale bars, 50 μm.

Results (Page 10)

We next administered the integrin α5β1-specific blocking peptide ATN-161 to WT mice (Supplemental Fig. 10a)...

Comment 2: siRNA – inadequate details of experimental conditions (amount of siRNA, timing etc)

Response: We apologize that our explanation of siRNA experiments was not clear. We included more detailed explanations in the methods as below.

Methods (Page 22)

siRNA transfection

For siRNA experiments, HUVECs were transfected with siRNAs targeting human CD36 (5'-AAGAGGAACTATATTG-TGCCTCCTGTCTC-3'), FATP3 (Santa Cruz Biotechnology, SASI_Hs01_00100092), FATP4 (Santa Cruz Biotechnology, SASI_Hs01_00047-530), ITGβ1 (5'-TGATAGATCCAATGGCTTA-3'), ITGα5 (Bioneer, 1075709), ITGαV (Bioneer, 1075799), ITGβ3 (Bioneer, 1075875), ITGβ5 (Bioneer, 1075906), TIE2 (5'-GGCUAGUAAGAUAUCAAUGGUdTdT-3'), NFATc1 (5'-CCCGUUCACGUCAGUUUCUACGUCU-3') or a scrambled control (5'-UAGCGACUAAACACAUCA-3') were used. Transfections of siRNA into the HUVECs were performed using Lipofectamine® RNAiMAX (Invitrogen, Waltham, Massachusetts) according to the manufacturer's protocol. Briefly, 20nM of siRNA was transfected with RNAiMAX and knockdown was assessed by RT-PCR. Experiments were conducted 48 hr after siRNA transfection.

Comment 3: HUVEC plated without gelatin or fibronectin – how good is adherence?

Response: We appreciate this comment. In order to minimize integrin activation by coating with ECM proteins such as gelatin or fibronectin (Davidenko et al., *J Mater Sci Mater Med.*, 2016), we used plates that were not coated with ECM proteins for this study. However, as the reviewer mentioned, this may indeed lead to poor adhesion of cells, so we employed a cell adhesion assay to compare adherence between dishes that were coated with different ECM. In our experimental condition, there were no difference in adherence of HUVECs plated with or without gelatin or fibronectin (Supplementary Fig. 6a). We included these additional findings and their descriptions into the revised manuscript accordingly as below.

Supplemental Fig.6

a Comparison of cell adhesion (OD) of HUVECs cultured on uncoated plates or those coated with 0.1% gelatin or 10 µg/mL fibronectin. Each dot indicates a value obtained from one well; $n = 6$ /group pooled from two independent experiments. Horizontal bars indicate mean \pm SD and P values versus Uncoated by two-tailed Student's t -test. NS, not significant.

Results (Page 7)

...we used uncoated plates to minimize integrin activation by ECM proteins such as gelatin or fibronectin³³; nonetheless, the adherence of HUVECs was comparable to coated dishes (Supplementary Fig. 6a).

Methods (Page 22)

Cell adhesion assay

For cell adhesion assays, 48-well tissue culture plates were either uncoated or coated with 0.1% gelatin or 10 $\mu\text{g}/\text{mL}$ fibronectin in PBS at 4°C for 1 h, air dried, and rinsed once with PBS. After being serum deprived at 37°C for 8 h, HUVECs were detached and washed twice, and plated onto the wells in serum-free medium containing 0.1% BSA at 5×10^4 per well. Cells in three wells of the quadruplicate were allowed to adhere to the coated/uncoated surface for 30 min, followed by four intensive washes to remove nonadherent cells, and incubated with 44 μM resazurin (#R7017, Sigma-Aldrich) in complete medium for 2 h. Resazurin fluorescence was then measured with a microplate reader (excitation 530 nm, emission 590 nm, cut-off 550 nm). Values were normalized to control.

Comment 4: N values – adipocyte size measured from only 3 mice per genotype – inadequate n value. Cell size data needs to be presented as an average per animal, rather than showing all individual counts. What are the area units? Why is the VAT adipocyte area less than 0? These numbers do not make sense – should be presented as adipocyte cross-sectional area. This is a major aspect of the study and thus data must be presented in a more rigorous and understandable way.

Response: This is a valid point that needs a more defined description. As described in the method that we have originally employed, Adiposoft (Galarraga et al., *J Lipid Res.*, 2012) provides arbitrary numbers that does not provide numerical area units. Therefore, as the reviewer suggested, we provide the actual value of cross-sectional area, and present data as average of 4-6 mice per group in the revised manuscript accordingly as shown in the example below.

Fig 1

e Representative H&E-stained images and comparisons of adipocyte size (diameter; μm) in different adipose tissues in WT and *Angpt2 Δ Ad* mice. $n = 4\sim 6$. Horizontal bars indicate mean \pm SD and P values versus WT by two-tailed Student's t -test. Scale bars, 50 μm .

Comment 5: All experiments conducted on male mice – given the differential role of SAT in male and female, it would be valuable to compared sexes.

Response: We appreciate this comment. To address this issue, we performed additional experiments to examine whether adipocyte-derived Angpt2 relates to sexual dimorphism in body-fat distribution. First, we analyzed adipose tissue weights and sizes of adipocytes in female(f)-*Angpt2*^{ΔAD} mice. We used the congenital Cre mouse, since inducible Cre needs tamoxifen-mediated Cre recombination which blocks estrogen receptors, thus affecting adipose tissue growth (Jeffery et al., *Cell Metab.*, 2016). We examined 24.0% less weight of SAT and 1.70-fold more weight in BAT in female f-*Angpt2*^{ΔAD} mice (Reviewer only Fig 2), which were comparable to 26.8% less SAT and 1.82-fold more BAT weight in male *Angpt2*^{ΔAD} mice (Supplementary Fig. 2). Thus, we were not able to examine significant differences in adipose tissue weights and sizes compared with the changes in male *Angpt2*^{ΔAD} mice. Therefore, we suspect that Angpt2-mediated endothelial FA uptake is not a differential factor in sexual dimorphism of body fat distribution in lean mice, however, its role in obesity is unclear. We have discussed about this important view raised by the reviewer in the revised manuscript as below.

Reviewer only Fig. 2

a Diagram for generation of female(f)-*Angpt2*^{ΔAd} mice, and their analyses at 8 weeks after birth.

b, c Comparison of body weight and fat weight in different adipose tissues between WT and f-*Angpt2*^{ΔAd} mice. Each dot indicates a value obtained from one mouse; *n* = 4 mice/group pooled from three independent experiments. Horizontal bars indicate mean ± SD and *P* values versus WT by two-tailed Student's t-test. NS, not significant.

d Representative H&E-stained images and comparisons of adipocyte size (diameter; μm) in different adipose tissues in WT and f-*Angpt2*^{ΔAd} mice. *n* = 4. Horizontal bars indicate mean ± SD and *P* values versus WT by two-tailed Student's t-test. Scale bars, 50 μm.

Discussion (Page 14)

Meanwhile, it is well known through clinical observation that subcutaneous obesity is more frequent in females and is less morbid than visceral obesity that is more frequent in males^{2,51}. Whether adipocyte-derived *Angpt2* and its role in endothelial fatty acid transport is involved in the sexual dimorphism of body fat distribution needs to be studied.

Comment 6: Normalization of RNA – relative expression was calculated by what procedure?

Response: We have now included more detailed information for RNA normalization into the methods as below.

Methods (Page 18)

Quantitative RT-PCR analyses

Quantitative real-time PCR was performed using FastStart SYBR Green Master mix (Roche) and QuantStudio3 (Applied Biosystems) with the indicated primers. The real-time PCR data were analyzed with QuantStudio Software (Applied Biosystems). Results were calculated using the delta delta CT method (Livak and Schmittgen, *Methods*, 2001), with *36b4* used for normalization of *in vivo* samples and *Gapdh* for normalization of *in vitro* samples. Primers for the quantitative real-time PCR are shown in Supplemental Table 3.

Comment 7: The timing of analysis of adipose tissue is different in every model (after 4 weeks for *Angpt2* deletion, after 1 week for b1-deletion, after 1-2 weeks for b1 blocking peptide). What is the rationale for these different time points?

Response: This is a valid point that needs to be addressed. In case of inducible *Angpt2* deletion, our main goal was to examine changes in adipocyte morphology and size. However, tamoxifen treatment at high doses is deleterious to adipocyte health, which requires additional two weeks after the last tamoxifen treatment for adipocytes to fully recover (Ye et al., *Mol Metab.*, 2015). This is the reason for the 4-

week regimen for inducible *Angpt2* model. In case of endothelial- *ITGβ1* deletion, 2-week or more regimen for tamoxifen induction results in adverse effects in blood vessel integrity whereas 1-week regimen still show changes in adipocyte morphology without affecting blood vessel integrity (Supplementary Fig. 9b-d) (we have addressed this issue in Comment 17 for Reviewer 1). Finally, for ATN-161 treatment, which specifically targets *ITGα5β1*, showed serial effects in a time-dependent manner in adipocyte morphology as shown in Fig 7b. Nevertheless, data regarding analysis of ATN-161 treatment was set at 1-week treatment to match with EC-specific *ITGβ1* depletion.

Data issues:

Comment 8: Characterization of vasculature inadequate (Fig. 1 j) – what is being assessed? Relative intensity is not explained – how does this relate to number of vessels/vessel density? It appears that capillary number is increased in the *Ang2* deficient mice. Since vascularization is intimately related to transport capacities of various tissues, it is not possible to make conclusions in the absence of a more thorough characterization of capillarization.

Response: This is a valid point that needs to be addressed. We have replaced the representative figures with a low-magnified view and analyzed vascular density and vessel area with our established method (Park et al., *Nat Commun.*, 2017; Kim et al., *J Clin Invest.*, 2017). Vascular density was measured as *CD31*⁺ vessel area divided by total measured area and presented as a percentage. Of note, we still detected no difference in vessel density between WT and *Angpt2*^{iΔAd} mice (Fig. 1j, k), which signifies that the changes in trans-endothelial FA uptake are not affected by vascularity itself. We included these additional findings and their descriptions into the revised manuscript accordingly as below.

Fig.1

j, k Representative images and comparison of vascular density and *CD31*⁺ vessel area in SAT of WT and *Angpt2*^{iΔAd} mice. Each dot indicates a value obtained from one mouse; *n* = 3 mice/group pooled from two independent experiments. Horizontal bars indicate mean ± SD and *P* values versus WT by two-tailed Student's *t*-test. NS, not significant. Scale bars, 100 μm.

Comment 9: Fig 2b Why does VAT not experience fatty acid uptake similar to BAT? This is not explained by the data presented. The authors should discuss possible reasons, as the bystander effect does not seem to apply here.

Response: This is a valid point that needs to be addressed. The difference between adipose depots responding to dietary fat intake could address this comment. While SAT expands through hypertrophy of existing adipocytes, VAT responds through hyperplasia (adipogenesis) upon dietary fat intake (Wang et al., *Nat Med.*, 2013; Jeffery et al., *Nat Cell Biol.*, 2015). This means that SAT expansion is mediated by fat intake itself whereas VAT expansion involves certain molecular cues to activate adipogenesis (Chau et al., *Nat Cell Biol.*, 2014; Ghaben and Scherer, *Nat Rev Mol Cell Biol.*, 2019). Expansion through adipogenesis in VAT clearly differs with compensatory FA uptake into BAT to burn excessive fat, or uptake of excessive lipid surge into skeletal muscle and liver, both involving FA uptake, not differentiation of newly generated adipocytes. We have discussed about this important point in the revised manuscript as below.

Discussion (Page 14)

The differences in adipose depots' response to dietary fat intake could explain why VAT does not exhibit compensatory fat uptake. While SAT expands through hypertrophy of existing adipocytes, VAT responds through hyperplasia of newly generated adipocytes upon dietary fat intake^{47,48}. This means that SAT expansion is mediated by fat intake itself whereas VAT expansion involves certain molecular cues to activate adipogenesis^{49,50}.

Comment 10: The data indicate low mRNA expression of Tie2 in SAT (Fig. 3) appear to be at odds with other studies. And would have major implications for any studies using Tie2-Cre deleter models to investigate SAT. Further documentation of the expression of Tie2 in the SAT is warranted. Furthermore, what is the level of Tie2 expression on the HUVEC used in the experiments?

Response: We appreciate this comment. In line with low mRNA expression in ECs of SAT (Fig. 3e), we confirmed low expression of Tie2 in *Tie2-eGFP* reporter mouse (Fig. 3g), and through Tie2 antibody staining (Supplementary Fig. 5b). We believe the essence of endothelial FA transport in SAT is through the organotypic characteristics of ECs in SAT that has unique expressions of angiocrine-receptors. In case of HUVECs that we used in our experiments, adequate expression of Tie2 was observed (Reviewer only Fig. 3a), which is the reason we had to activate integrin motifs to enable Angpt2 to bind to integrin receptors over Tie2. In line with this, Angpt2 treatment itself was able to maximize FA uptake in siTie2 HUVECs (Supplementary Fig. 6c and Reviewer only Fig. 3b) or primary cultured ECs of SAT (Supplementary Fig. 7). We included these additional findings and their descriptions and discussed this in the revised manuscript accordingly as below.

Fig.3

g Representative images of Tie2 expression in SAT and BAT of Tie2-GFP mice. Magnified views of dotted-line box are shown in lower panels. Scale bars, 50 μ m.

Supplementary Fig.5

b Representative images of Tie2 (antibody) expression in SAT of WT mice. Scale bars, 30 μ m.

Results (Page 7)

Immunohistochemistry analyses supported these reciprocal expression levels of integrin $\alpha 5\beta 1$ and Tie2 in the ECs of SAT, which clearly differs with other depots (Fig. 3f, g and Supplementary Fig. 5a,b).

Reviewer only Fig. 3

a Immunoblot analysis of indicated proteins in HUVEC.

b Comparisons of Tie2 mRNA expression in siCtrl and siTie2 HUVECs. Each dot indicates a value obtained from experiment; n = 3/group. Horizontal bars indicate mean \pm SD and P values versus WT by two-tailed Student's t-test.

Discussion (Page 14)

This selectivity can be explained by the distribution of its receptors; ECs in SAT highly express ITG α 5 β 1 but barely express Tie2, whereas ECs in VAT behave in the opposite fashion. EC-specific integrin KO mice phenocopied adipocyte-specific Angpt2 KO mice further confirming that the integrin but not Tie2 is a key determinant of Angpt2 action on endothelial FA uptake in SAT.

Comment 11: The Ribotag strategy is innovative. Analysis of these samples for RNAseq hardly justified since only 3 target genes are shown. Methodology does not adequately explain how the RNAseq data are displayed...what have they been normalized to?

Response: We apologize for inadequate details about methodology regarding the RiboTag strategy. We have now included more detailed information for the methodology and normalization methods regarding Ribotag-based RNA-Seq analysis as below.

Methods (Page 24)

RNA-seq analysis of RiboTag-captured mRNA

For RNA-seq data analysis, four to five biological replicates of mRNA isolated by RiboTag method were used for analysis. The normalized count values were processed based on Quantile normalization method using the Genowiz™ version 4.0.5.6 (Ocimum Biosolutions) and used for heatmaps and bioinformatics analysis. RNA-Seq gene expression heatmap was generated with Multiple Experiment Viewer (MeV) from The Institute of Genomic Research (TIGR). The indicated expression ratio in a heatmap reflects the normalized count of each replicate apart from the mean gene expression value over all condition.

Comment 12: Does Angpt2 deletion affect integrin expression in EC, or vice versa? Does HF diet impact integrin a5b1 expression?

Response: To address these comments, we examined ITG α 5 β 1 expression in *Angpt2*^{i Δ Ad} mice and WT mice treated with high fat diet. Of note, expression of ITG α 5 β 1 was decreased in *Angpt2*^{i Δ Ad} mice (Reviewer only Fig. 4a), whereas high fat diet treatment enhanced ITG α 5 β 1 expression in ECs of SAT compared with normal chow diet treated mice (Reviewer only Fig. 4b). These data implicate that adipocyte-derived Angpt2 regulates ITG α 5 β 1 expression. However, we did not exhibit any changes in Angpt2 expression in SAT of *ITG β 1* ^{Δ EC} mice compared to WT mice (Reviewer only Fig. 4c), signifying that the receptor expression itself does not regulate its ligand expression.

Reviewer only Fig. 4

a Representative images of ITG α 5 β 1 expression in SAT of WT and *Angpt2* ^{Δ Ad} mice. Scale bars, 50 μ m.

b Representative images of ITG α 5 β 1 expression in SAT of WT mice fed with normal chow diet (ND) or high fat diet for 7 days (HFD-7d). Scale bars, 50 μ m.

c Comparison of *Angpt2* mRNA expression in SAT of WT and *ITG β 1* ^{Δ EC} mice. Each dot indicates a value obtained from one mouse and n = 3 mice/group. Horizontal bars indicate mean \pm SD and P values versus WT by two-tailed Student's t-test.

Comment 13: In vitro experiments would have been strengthened by using a microvascular EC from adipose tissue (since vascular bed-specific endothelial cell phenotype is well established) instead of HUVEC.

In order to address this comment, we have cultured primary endothelial cells from SAT and compared the effect of *Angpt2* with primary endothelial cells from VAT. First, we employed a previously published method in culturing primary endothelial cells of murine adipose tissues (Kajimoto et al., *J Immunol Methods*, 2010), and validated 92.7% purity of CD31⁺ fraction (Supplementary Fig. 7a-c). Next, we compared the effects of *Angpt2* treatment with or without Mn²⁺ addition in primary endothelial cells from SAT and VAT (Supplementary Fig. 7d). Of note, *Angpt2* treatment alone enhanced FA uptake in time- and dose-dependent manner only in SAT ECs (Supplementary Fig. 7e-f). Importantly, this effect was masked by ITG α 5 β 1 antagonist (ATN-161) treatment (Supplementary Fig. 7g). As *Angpt2* treatment alone without the aid of integrin motif activation enabled enhancement of FA uptake, this implies that *in vitro* culture mimics the organotypic characteristics of ECs of SAT. We included these additional findings and their descriptions into the revised manuscript accordingly as below.

Supplemental Fig 7. Angpt2 stimulates FA uptake through integrin $\alpha 5\beta 1$ signaling in primary cultured ECs from SAT

a-c Diagram depicting the FA uptake of primary cultured ECs from adipose tissues. Primary cultured ECs from WT mice were validated for CD31⁺ expression. **d** Comparison of FA (BODIPY C-12, 8 μ M) uptake after treatment with vehicle, Mn²⁺ (1 mM), Angpt2 (2.5 μ g/ml) or Mn²⁺ (1 mM) + Angpt2 (2.5 μ g/ml) for 15 min. **e** Comparisons of FA (BODIPY C-12, 8 μ M) uptake after treatment with Angpt2 (2.5 μ g/ml) for indicated time points and . n = 5 for each group. Horizontal bars indicate mean \pm SEM and *** < 0.001 versus VAT EC by two-tailed Student's t-test. **f** Comparisons of FA (BODIPY C-12, 8 μ M) uptake after treatment with indicated concentration of Angpt2. n = 5 for each group. Horizontal bars indicate mean \pm SEM and ***P < 0.001 versus VAT EC by two-tailed Student's t-test. **g** Comparisons of FA (BODIPY C-12, 8 μ M) uptake after treatment with or without Angpt2 (2.5 μ g/ml), ITG $\alpha 5\beta 1$ blocking peptide (ATN-161, 10 μ M) for 15 min. Each dot indicates a mean of triplicate values from two independent experiments. Horizontal bars indicate mean \pm SD and P values versus Angpt2 by two-tailed Student's t-test.

Results (Page 8)

To strengthen our claim that Angpt2 induces organotypic FA uptake in SAT ECs, we compared the effect of Angpt2 on primary endothelial cells from SAT and VAT (Supplementary Fig. 7a). First, we employed a previously published method for culturing primary endothelial cells of murine organs³⁷, and validated its 92.7%

(Supplementary Fig. 7a-c). Next, we compared the effects of Angpt2 treatment with or without Mn^{2+} in primary endothelial cells from SAT and VAT (Supplementary Fig. 7d). Of note, Angpt2 treatment alone enhanced FA uptake in time- and dose-dependent manners only in SAT ECs (Supplementary Fig. 7d-f). Importantly, this effect was masked by ATN-161 treatment (Supplementary Fig. 7g). These data demonstrate that the endothelial integrin $\alpha 5\beta 1$ in SAT mediates Angpt2-induced FA uptake.

Methods (Page 20)

Cell culture

For primary culture of endothelial cells from SAT and VAT, we employed and modified a previously published method for culturing primary endothelial cells of murine adipose tissues³⁷. Dissected SAT and VAT were chopped with scissors and incubated in digestion medium containing 0.2% collagenase type 2 (Worthington) for 30 min at 37°C on a shaker. The resulting suspension was lysed with ACK Lysing Buffer (Life Technologies) for 5 min at 37°C and centrifuged at $470 \times g$ for 5 min. After several washes, cell suspension was filtered through 100 μm nylon cell strainer and washed. To enrich the endothelial cell fraction, cells were incubated for 20 min with anti-CD31 Microbeads (Miltenyi) and selected using AutoMACS (Miltenyi) according to the manufacturer's instructions. In order to maximize endothelial cell survival and culture, cells were incubated with EGM2 containing 10% fetal bovine serum and 400 ng/ml of endothelial cell growth supplement (ECGS; Sigma). Culture-expanded monolayer of ECs under single passage that were validated to be CD31⁺ were used for all experiments.

Comment 14: Regarding the in vitro data showing CD36, integrin $\alpha 5\beta 1$ interaction (Fig.5): Is there evidence that these molecules remain on the cell surface? They appear to be internalized into vesicles. What is the duration of time of this association following Angpt2 stimulation? Is CD36 on the cell surface initially? Why does siCD36 not affect basal FFA uptake?

First, to address the localization of CD36 and integrin complex, we used cell surface markers to confirm that that this complex reside in the VE-cadherin⁺ plasma membrane (Supplementary Fig. 8a). In line with our response to the first remark by the author regarding luminal/abluminal localization of integrins and CD36, we provide evidence that abluminal ITG $\beta 1$ binds to luminal CD36 through transmembrane lipid rafts (Caveolin-1) (Supplementary Fig. 8b). Since it is well known that ITG $\beta 1$ stabilize lipid rafts (Bodin et al., *J Cell Sci.*, 2005), and that lipid rafts interact with CD36 in the plasma membrane (Tran et al., *J Biol Chem.*, 2011), we speculate that ITG $\beta 1$ -lipid raft-CD36 could form a complex in the plasma membrane.

It is the duration and FA dose that matters in our experimental settings. After 8 hours of serum starvation, the FA uptake and CD36 localization effect by Angpt2 is observed in less than 5 minutes with a low dose of added FA (Fig 4c,d). At this condition, siCD36 treated ECs are unaffected, since CD36 is not initially localized on the surface at serum starved ECs (Fig 4i).

We speculate that the rapid action of Angpt2 makes it possible to respond to acute postprandial surge of plasma lipid, as trans-endothelial FA transport in SAT enables buffering of excessive plasma lipid into SAT. We included these additional findings and discussed about this important issue in the revised manuscript as below.

Supplementary Fig.8

a Representative images of ITGβ1, CD36, and VE-cadherin complex (arrowheads) in HUVECs treated with Mn²⁺ (1 mM) + Angpt2 (2.5 ug/ml) for 15 min. Magnified view is shown in right panels. Scale bars, 10 μm.

b Representative images of ITGβ1, CD36, and Caveolin-1 complex (arrowheads) in HUVECs treated with Mn²⁺ (1 mM) + Angpt2 (2.5 ug/ml) for 15 min. Magnified view is shown in right panels. Scale bars, 10 μm.

Results (Page 9)

Since ITGβ1 faces the basolateral side of ECs³⁸, and CD36 is located in the luminal side of ECs³⁹, we speculated that trans-membrane lipid rafts, which interact with both molecules⁴⁰, could mediate this binding complex. Indeed, we detected co-localization of this complex with Caveolin-1⁺ lipid rafts in plasma membrane of ECs (Supplementary Fig. 8a,b).

Discussion (Page 13)

Thus, fat intake-induced Angpt2 released from adipocytes can stimulate FA uptake by neighboring ECs without systemic impact. In eliciting this outcome, integrin receptors face the basolateral side to bind with the extracellular matrix⁴⁶. Although ITGα5β1 is on the basolateral side and CD36 resides on the luminal side, these seem to form a complex through transmembrane lipid rafts^{39,40}. Thus, Angpt2 produced from the adipocytes could aggregate more easily than Angpt2 released from ECs circulating inside the lumen.

Comment 15: A previous paper (An et al. eLife 2017) shows decreased *angpt2* expression with HFD on SAT whereas this study shows that HFD increases *angpt2* with short (3-7 days) and longer (3 mo) periods. The authors should discuss this discrepancy. Furthermore, An et al. (2017) have already shown that inactivation of *Angpt2* exacerbates HFD induced metabolic defects and impairs lipid disposal, which was associated with TG accumulation in the liver.

Response: This is a valid point. First of all, in the referenced paper, An et al. gave the mice high fat diet for 6 months and found that *Angpt2* is decreased, which results in morbidly obese phenotype. On the other hand, we made our observation at 3-7 days and 3 months after high fat diet, which led to increased levels of *Angpt2* compared with basal; this is also the time point that which we propose positive and physiological activation of *Angpt2* occurs to cope with excessive lipid. Secondly, the reference paper used *Angpt2*-blocking antibody, which systemically blocks all *Angpt2*-mediated interactions and therefore *Tie2*-mediated angiogenesis as well as our *Angpt2*-*ITGα5β1* interaction. On the other hand, we specifically knocked out *Angpt2* in adipocytes to block the interaction between *Angpt2* and basolateral *ITGα5β1*. As the reviewer mentioned in comment 8, angiogenesis itself could affect FA intake, and thus result in adverse metabolic defects. To support this we have some previous data that we did not include into the manuscript. Briefly, when we overexpressed COMP-Ang1 (a designed Angiopoietin-1 variant to target *Tie2*; Cho et al., *Proc Natl Acad Sci U S A*, 2004) to stimulate *Tie2*-mediated angiogenesis, which is the opposite effect of the referenced paper, we observed metabolic improvements such as enhanced serum-lipid disposal and less ectopic lipid accumulation (Reviewer only Fig. 5). Therefore we are not contradicting the previously published work, but we believe the difference between An et al. and our work mainly lies in 2 features: time point and specificity of deletion and therefore the isolated disruption of *Angpt2*-*ITGα5β1* interaction without disturbing *Angpt2*-*Tie2*-mediated angiogenesis.

Reviewer only Fig. 5

a, 8-week-old control and Ang1^{OE} mice were analyzed.

b, Comparisons of blood vessel density in SAT and VAT. Each group, n = 3 to 5. Values are mean ± SD. The p values were determined by the unpaired Student's t-test. *, p < 0.05 vs. Control mice.

c-l, 8-week-old control and Ang1^{OE} mice were fed with high-fat diet for 12 weeks. Each group, n = 5 to 7. Values are mean ± SD. The p values were determined by the unpaired Student's t-test. *, p < 0.05 vs. Control mice. **(c)** Body weight changes after high-fat diet. **(d, e)** Comparisons of serum levels of TG and FFA. **(f)** Representative images of various H&E-stained tissues. **(g-j)** Comparisons of liver-

to-body weight ratio and serum levels of ALT, AST, and cholesterol. (k) Intraperitoneal glucose tolerance test. (l) Intraperitoneal insulin tolerance test. m-n, Mice fed with high-fat diet for 12 weeks were subjected to metabolic cage assessment with CLAMS analysis. Each group, n = 3 to 4. Values are mean ± SD. The p values were determined by the unpaired Student's t-test. *, p < 0.05 vs. Control mice. Comparisons of O₂ consumption (m) and CO₂ production (n).

Comment 16: Data for Human geoset analyses is poorly described, making it difficult to draw conclusions. What is the relevance of values shown in Fig.8g.

Response: We apologize that our explanation on the Human geoset analyses in Fig. 8g was not clear. We intended to highlight that the expression of ITG α 5 receptor was enriched in non-diabetic obese individuals compared with lean individuals (left panel), and that the expression was enriched in SAT compared with VAT in non-diabetic obese individuals (right panel). We have revised the results with additional explanation as below to fully describe the data in the revised manuscript.

Results (Page 11)

By comparison analysis of human GEO data, we confirmed enhanced expression of integrin α 5 in SAT compared with VAT in NDO patients (Fig. 8g). Likewise, we confirmed enriched expression of ITG α 5 β 1 in endothelial cells of SAT of NDO patient (Supplementary Figure 11d). Thus, integrin receptor expression is enriched in endothelial cells of SAT in non-diabetic obese individuals.

Comment 17: Previous studies using endothelial-specific depletion of beta1 integrin have noted severe permeability/vascular integrity issues. An in vivo assessment of vascular permeability is necessary to document the extent to which permeability might be altered in these mice.

Response: This is a valid point that needs to be addressed. In order to address this comment, we assessed vascular permeability in *ITG β 1^{i Δ EC}* mice by employing dextran injection method (Park et al., *Cancer Cell*, 2016). Compared with *ITG β 1^{i Δ EC}* mice observed after 1-week of tamoxifen induction, the extent of vascular permeability was similar to WT, whereas *ITG β 1^{i Δ EC}* mice observed after 2-weeks of tamoxifen induction showed severe permeability defects (Supplementary Fig. 9c). Moreover, Ter119⁺ red blood cells were intact in blood vessels in *ITG β 1^{i Δ EC}* (1wk) mice (Supplementary Fig. 9b). Therefore, the tamoxifen regimen we used in this study for *ITG β 1^{i Δ EC}* mice is well-suited to study the changes in adipose morphology without adverse effects in vascular integrity. We now described the results with additional explanation as below to fully describe the data in the revised manuscript.

Supplemental Fig 9. Preserved vascular integrity and density in *ITGB1*^{ΔEC} mice through 1-week tamoxifen regimen

b Representative images of Ter119 expression in SAT and VAT in *ITGB1*^{ΔEC} (1wk) mice. Scale bars, 50 μm.

c Representative images and comparison of vascular permeability in ECs of SAT after FITC-dextran injection in WT and *ITGB1*^{ΔEC} mice with different tamoxifen regimen. Each dot indicates a value obtained from one mouse and n = 3 mice/group. Horizontal bars indicate mean ± SD and P values versus WT by two-tailed Student's t-test. Scale bars, 50 μm.

Results (Page 9)

There were no changes in vascular integrity or density in *ITGB1*^{ΔEC} mice after 1 week of tamoxifen treatment (Supplementary Fig. 9a-d).

Methods (Page 23)

In vivo vascular permeability assay

Vascular leakage was analyzed after i.v. injection of 100 μl of FITC-conjugated dextran (4 mg/ml, 70 kDa, Sigma-Aldrich) 5 min before sacrifice. Mice were anesthetized and perfused by intracardiac injection of 1% PFA to remove circulating dextran.

Writing issues:

Comment 18: Many parts of the results are inadequately described within the text.

Response: We apologize for the inadequate description of the results in the manuscript. Following the reviewer's suggestion, we revised the whole manuscript by adding, rewording, rephrasing and modifying the text with adequate description of each data throughout the revision. In particular, we provided additional information regarding human genetic datasets, difference in receptors expression, vascular permeability and so on.

Comment 19: The discussion is inadequate in depth and scope.

Response: Following the reviewer's suggestion, we revised the discussion to be more focused on the organotypic characteristics of blood vessels in the adipose tissue and how Angpt2-ITGa5 β 1 is deeply involved in this system as marked by blue in the revised discussion.

***References for responses to comments of Reviewer #1**

Bodin, S. et al. Integrin-dependent interaction of lipid rafts with the actin cytoskeleton in activated human platelets. *J Cell Sci* 118, 759-769, doi:10.1242/jcs.01648 (2005).

Chau, YY. et al. (2014). Visceral and subcutaneous fat have different origins and evidence supports a mesothelial source. *Nat Cell Biol.*, Apr; 16(4): 367–375.

Cho CH. et al. (2004). COMP-Ang1: a designed angiopoietin-1 variant with nonleaky angiogenic activity. *Proc Natl Acad Sci U S A*. Apr 13;101(15):5547-52.

Galarraga, M. et al. (2012). Adiposoft: automated software for the analysis of white adipose tissue cellularity in histological sections. *J Lipid Res*. 53, 2791-2796.

Jeffery E, Church CD, Holtrup B, Colman L, Rodeheffer MS. (2015). Rapid depot-specific activation of adipocyte precursor cells at the onset of obesity. *Nat Cell Biol.*, Apr;17(4):376-85.

Jeffery E et al., (2016). The Adipose Tissue Microenvironment Regulates Depot-Specific Adipogenesis in Obesity. *Cell Metab*. Jul 12;24(1):142-50.

Kajimoto K et al., (2010). Isolation and culture of microvascular endothelial cells from murine inguinal and epididymal adipose tissues. *J Immunol Methods*. 2010 May 31;357(1-2):43-50.

Kim J et al., (2017). YAP/TAZ regulates sprouting angiogenesis and vascular barrier maturation. *J Clin Invest*. Sep 1;127(9):3441-3461.

Livak KJ, Schmittgen TD. (2001). Analysis of relative gene expression data using real-time quantitative PCR and the 2(-Delta Delta C(T)) Method. *Methods*. Dec;25(4):402-8.

Minami T et al., (2013). The calcineurin-NFAT-angiopoietin-2 signaling axis in lung endothelium is critical for the establishment of lung metastases. *Cell Rep*. Aug 29;4(4):709-23.

Ojakian, G. K. & Schwimmer, R. Regulation of epithelial cell surface polarity reversal by beta 1 integrins. *J Cell Sci* 107 (Pt 3), 561-576 (1994).

Park DY et al., (2017). Plastic roles of pericytes in the blood-retinal barrier. *Nat Commun.* May 16;8:15296.

Pitter, B., Werner, A. C. & Montanez, E. Parvins Are Required for Endothelial Cell-Cell Junctions and Cell Polarity During Embryonic Blood Vessel Formation. *Arterioscler Thromb Vasc Biol* 38, 1147-1158, doi:10.1161/ATVBAHA.118.310840 (2018).

Schoettl T, Fischer IP, Ussar S. (2018). Heterogeneity of adipose tissue in development and metabolic function. *J Exp Biol.* Mar 7;221.

Shah MS, Brownlee M, (2016). Molecular and Cellular Mechanisms of Cardiovascular Disorders in Diabetes. *Circ Res.* May 27;118(11):1808-29.

Strilic, B. et al. The molecular basis of vascular lumen formation in the developing mouse aorta. *Dev Cell* 17, 505-515, doi:10.1016/j.devcel.2009.08.011 (2009).

Tran, T. T. et al. Luminal lipid regulates CD36 levels and downstream signaling to stimulate chylomicron synthesis. *J Biol Chem* 286, 25201-25210, doi:10.1074/jbc.M111.233551 (2011).

Wang QA, Tao C, Gupta RK, Scherer PE, (2013). Tracking adipogenesis during white adipose tissue development, expansion and regeneration. *Nat Med.* Oct;19(10):1338-44.

Ye R et al., (2015). Impact of tamoxifen on adipocyte lineage tracing: Inducer of adipogenesis and prolonged nuclear translocation of Cre recombinase. *Mol Metab.* Aug 29;4(11):771-8.

Reviewer #2:

(Remarks to the Author)

This very interesting manuscript shows for the first time that angiopoietin-2 controls uptake of fatty acids into subcutaneous but not visceral adipose tissue. This is achieved by signaling through integrins on endothelial cells which subsequently promote proper localization of fatty acid transport proteins, thereby facilitating endothelial transport of fatty acids from blood plasma to adipocytes. The authors show that there is differential Angpt2 expression in obese vs. lean subjects and that this may contribute to insulin resistance. The manuscript is very well written, succinct and clear. The Material and Methods are well described. The figures are of high quality. If revised properly, this work will significantly advance the field. It demonstrates a key example of organ-specific angiocrine signaling that controls metabolism.

Response: We appreciate these encouraging comments. We hope that the reviewer would be satisfied with our responses to the following specific issues.

Major points:

Comment 1: The authors demonstrate nicely increased Angpt2 in a high fat diet model. Is there also a physiological relevance of this study? One should determine Angpt2 and its receptor levels in adipose tissue upon feeding/fasting cycles and whether this affects expression of CD36 and FATP3.

Response: We appreciate this constructive comment. In order to address this comment, we performed additional experiments to explore the changes in Angpt2 and its receptors during feeding/fasting cycles. First, mRNA expression of Angpt2 was decreased during fasting in SAT (Supplementary Fig. 11e), indicating that feed/fast cycles regulate Angpt2 expression. Accordingly, we observed similar changes in both ITG α 5 β 1 and CD36 expression in ECs of SAT (Supplementary Fig. 11f,g). In case of FATP3, we tried several commercial antibodies but none of them were suitable for immunofluorescence staining. We were not able to measure the changes in FATP3 mRNA expression since FATP3 is regulated in a post-translational manner (Jang et al., *Nat Med.*, 2016). We would need to wait until a better method such as animal models with modifications in FATP3 to measure its activity. We included these additional findings and discussed about this issue in the revised manuscript as below.

Supplementary Fig. 11

e Comparisons of Angpt2 mRNA expression in SAT during fed/fast cycle in WT mice.

Each dot indicates a value obtained from one mouse and $n = 4$ mice/group. Horizontal bars indicate mean \pm SD and P values versus WT by two-tailed Student's t-test.

f Representative images of ITG α 5 β 1 expression in SAT during fed/fast cycle in WT mice. Scale bars, 50 μ m.

g Representative images of CD36 expression in SAT during fed/fast cycle in WT mice. Scale bars, 50 μ m.

Results (Page 11)

Next, we observed changes in Angpt2 and its receptors during fast/fed cycle. Of note, Angpt2 expression was reduced by 58.8% in fasted mouse (Supplementary Fig. 11e). Consistently, expressions of both ITG α 5 β 1 and CD36 were downregulated in ECs during fasting (Supplementary Fig. 11f,g). These indicate that expressions of Angpt2 and its mediators in FA transport are reduced during energy output.

Discussion (Page 14)

On the other hand, genetic mouse models for FATPs and their phenotypes regarding endothelial fat metabolism have not yet been reported. Thus, studies using animal models with genetic modifications in different FATPs in ECs will be useful to demonstrate their importance in regulating endothelial FA transport.

Comment 2: Fig 2A and Fig 6C: how is this fatty acid uptake assay controlled? Do the authors measure the initial uptake into the blood stream that should be equal in both groups? How is the FA uptake into liver?

Response: This is a valid point that needs to be addressed. To measure tissue fatty acid uptake, we employed a previously established method (Khalifeh-Soltani et al., *Nat Med.*, 2014). Injected amount of FA were normalized to body weight of each mice, and plasma uptake of FA was measured and results were comparable (Supplemental Fig. 4a and 9h). In case of outliers, we ruled out mice that showed insufficient plasma FA uptake. As the reviewer mentioned, we measured other metabolic organs such as liver during our experiments, and the results were also comparable in non-obese mice (Supplemental Fig. 4b and 9i). We speculate that changes in uptake into organs such as liver would need prolonged HFD treatment. We included these additional findings and included detailed information regarding

the *in vivo* fatty acid uptake experiments in the revised manuscript as shown in the example below.

Supplemental Fig. 4

a, b Comparisons of ¹⁴C-oleic acid uptake in plasma and liver in WT and *Angpt2^{iAd}* mice. Each dot indicates a value obtained from one mouse; $n = 6\sim 8$ mice/group pooled from two independent experiments. Horizontal bars indicate mean \pm SD and P values versus WT by two-tailed Student's t -test. NS, not significant.

Supplemental Fig. 9

h, i Comparisons of FA (BODIPY C-16, 0.5 g/kg) uptake into plasma and liver between WT and *ITGB1^{ΔEC}* mice. Each dot indicates a mean of quadruplicate values from three independent experiments. Horizontal bars indicate mean \pm SD and P value versus vehicle by two-tailed Student's t -test. NS, not significant.

Methods (Page 19)

Fatty acid uptake *in vivo*

To measure FA absorption *in vivo* using fluorescent-dyes, we revised a previously described method²⁰. In brief, the appropriate amount of BODIPY fluorescent-conjugated FAs (BODIPY FL C16 and BODIPY 558/568 C12, Invitrogen) was calculated (0.5mg/kg) for each mice and dissolved in control solution (HBSS supplemented with 20 mM HEPES and 0.2% FFA free BSA) in a total volume of 200 μ l. Injection was performed intravenously, and tissues were harvested and snap-frozen at indicated time points. Tissues were then homogenized in RIPA buffer, centrifuged and supernatant was used. Fluorescence intensity was measured ($\lambda_{ex} = 485/\lambda_{em} = 520$ nm) in black, 96-well flat-bottom plates using a fluorescence microplate reader (Tecan). We subtracted the fluorescence signal of each tissue from mice treated with control solution, and normalized by the weight of the extracted tissue. For both methods, those with insufficient FA intake in plasma were ruled out for data analysis.

Comment 3: Is fatty acid production by the liver affected by *Angpt2*? In addition, what are the levels of triglycerides and free fatty acid in plasma under normal chow conditions?

Response: This is a valid point. In order to address this comment, we analyzed changes in mRNA expression of lipogenesis markers in liver and measured plasma levels of triglycerides and free fatty acid under normal chow conditions. Of note, there were neither changes in lipogenesis markers such as *Acc*, *Srebp1c*, *Fas*, *Chrebp*, and *Scd1* (Supplemental Fig. 4c) nor difference in plasma levels of triglycerides and free fatty acid between WT and *Angpt2*^{iΔAd} mice (Supplemental Fig. 4d,e). We included these additional findings and their descriptions into the revised manuscript accordingly as below.

Supplemental Fig. 4

c Comparisons of indicated mRNA expression in SAT between WT and *Angpt2*^{iΔAd} mice under normal chow diet. Each dot indicates a value obtained from one mouse; n = 3 mice/group. Horizontal bars indicate mean ± SD and P values versus WT by two-tailed Student's t-test. NS, not significant.

d, e Comparisons of plasma triglyceride (TG) and free fatty acid (FFA) levels between WT and *Angpt2*^{iΔAd} mice under normal chow diet. Each dot indicates a value obtained from one mouse; n = 4 mice/group pooled from two independent experiments

Comment 4: Fatty acid uptake is also under control of insulin. What are insulin and glucose levels under basal conditions? Are there any changes in pancreas morphology.

Response: This is a valid point. In order to address this comment, we analyzed plasma insulin and glucose levels under basal conditions. Of note, there were no differences in plasma glucose or insulin levels between WT and *Angpt2*^{iΔAd} mice (Supplemental Fig. 4f,g). Moreover, immunofluorescence analysis of pancreas revealed no differences in glucagon and insulin expression (Supplemental Fig. 4h).

Further analysis of insulin and glucose tolerance tests revealed no change in insulin sensitivity (Supplemental Fig. 4i,j). Thus, these findings imply that Angpt2-mediated FA uptake is not primarily due to changes in insulin levels. We included these additional findings and their descriptions into the revised manuscript accordingly as below.

Supplemental Fig. 4

f, g Comparison of plasma insulin and glucose levels between WT and *Angpt2^{iΔAd}* mice under normal chow diet. Each dot indicates a value obtained from one mouse; n = 7(insulin), n = 4(glucose) mice/group pooled from two independent experiments. Horizontal bars indicate mean \pm SD and P values versus WT by two-tailed Student's t-test. NS, not significant.

h Representative images of indicated proteins in pancreas of WT and *Angpt2^{iΔAd}* mice under normal chow diet. Scale bars, 100 μ m.

i, j Comparisons of intraperitoneal glucose tolerance test (IPGTT) and intraperitoneal insulin tolerance test (IPITT) between WT and *Angpt2^{iΔAd}* mice under normal chow diet. Each dot indicates a value obtained from one mouse; n = 6 mice/group pooled from three independent experiments. Horizontal bars indicate mean \pm SEM and P values versus WT by two-tailed Student's t-test.

Results (Page 11)

Importantly, fatty acid production, uptake by liver, or systemic levels of insulin or glucose, which are regulators of fatty acid mobilization, were unchanged, indicating direct action of Angpt2 on FA uptake (Supplementary Fig. 4b-j).

Comment 5: A striking feature of this study is the clear organ specificity of Angpt2 action what the authors can convincingly correlate with the differential mRNA expression patterns of the integrin receptors. It would be helpful to

show the differential protein expression also. Therefore integrin staining should not only be shown for SAT (fig. 3G) but also VAT and BAT.

Response: We appreciate this insightful comment. Following the reviewer's suggestion, we performed additional experiments to compare the expression of ITG α 5 β 1 in various adipose depots. Of note, ITG α 5 β 1 expression was preferentially enriched in ECs of SAT, but not in VAT or BAT (Fig. 3f). Thus, these findings imply the organotypic expression of ITG α 5 β 1 among adipose depots. We included these additional findings and their descriptions into the revised manuscript accordingly as below.

Fig 3

f Representative images of ITG α 5 β 1 expression in various adipose tissues of WT mice. Scale bars, 50 μ m.

Results (Page 7)

Immunohistochemistry analyses supported these reciprocal expression levels of integrin α 5 β 1 and Tie2 in the ECs of SAT, which clearly differs with other depots (Fig. 3f, g and Supplementary Fig. 5a, b).

Comment 6: Another striking aspect of this study is that only Angpt2 derived from adipocytes but not from endothelial cells is important for FA transport? The authors speculate that endothelial release of Angpt2 occurs into the blood stream whereas adipocytes release Angpt2 to the basolateral membrane of endothelial cells. This is intriguing. It would be very interesting to see where the integrin receptors are located on endothelial cells (apical or basolateral) to substantiate this.

Response: We appreciate this insightful comment. Following the reviewer's suggestion, we performed additional experiments to locate the expression of

ITG α 5 β 1 on endothelial cell surface. By employing previous methods on specifying apical and basolateral sides of endothelial cells (Strilić et al., *Dev Cell.*, 2009; Pitter et al., *Arterioscler Thromb Vasc Biol.*, 2018), we observed ITG α 5 β 1 expression mainly on collagen IV⁺ basolateral membrane, but not on CD34⁺ apical membrane of endothelial cells (Supplemental Fig. 8c,d). We included these additional findings and their descriptions into the revised manuscript accordingly as below.

Supplemental Fig. 8

c, d Representative images of ITG α 5 β 1 expression on Collagen IV⁺ basolateral membrane (arrowheads) but not on CD34⁺ apical membrane (arrowheads) of endothelial cells in SAT of WT mice. Magnified view is shown in right panels. Scale bars, 20 μ m.

Results (Page 9)

In line with this finding, we found that ITG α 5 β 1 is mainly located on Collagen IV⁺ CD34⁻ basolateral membrane of ECs^{41,42} in SAT (Supplementary Fig. 8c, d).

Comment 7: The changes in BAT are not convincingly explained. Are there any indications for altered thermogenesis?

Response: This is a valid point that needs to be addressed, and please note that this issue was also discussed in Reviewer 3's comment 1. As the reviewer requested, we performed additional experiments to examine alterations in thermogenesis of BAT in *Angpt2*^{i Δ Ad} mice. We first examined changes in thermogenic markers and mitochondrial morphology in BAT of *Angpt2*^{i Δ Ad} mice. Of note, thermogenic markers such as *Ucp1*, *Dio2*, *Pgc1 α* , *Cidea*, and *Elovl3* were significantly decreased in BAT of *Angpt2*^{i Δ Ad} mice (Supplemental Fig. 12c). Moreover, cristae of mitochondria were unpacked and filled with vacuoles (Supplemental Fig. 12d), altogether indicating defects in thermogenesis (Nam M et al., *Sci Rep.*, 2017). We included these additional results and their descriptions into the revised manuscript accordingly as below.

Supplemental Fig 12

c Comparisons of indicated mRNA expression in BAT between WT and *Angpt2^{iΔAd}* mice. Each dot indicates a value obtained from one mouse; n = 5~6 mice/group. Horizontal bars indicate mean ± SD and P values versus WT by two-tailed Student's t-test.

d Representative electron microspore images showing mitochondrial morphology in BAT of WT and *Angpt2^{iΔAd}* mice. Magnified view is shown in right panels. Scale bars, 2 μm.

Results (Page 11)

...while thermogenic markers were downregulated in BAT in *Angpt2^{iΔAd}* mice (Supplementary Fig 12c). Likewise, electron microscope analysis revealed unpacked cristae and vacuole filled mitochondria in BAT of *Angpt2^{iΔAd}* mice (Supplementary Fig. 12d).

Comment 8: The authors show immunohistochemistry of UCP1 in WAT. Is this sufficient to rule out quantitative changes? Western blot might be more appropriate. Also O₂ consumption of isolated WAT would be helpful for better interpretation.

Response: Thank you for this constructive comment. As the reviewer suggested, we quantified UCP1 protein in SAT by western blot, but could neither detect any signal (upper band compared to BAT control) nor difference between WT and *Angpt2^{iΔAd}* samples (Supplemental Fig. 3a). In order to address the next comment, we isolated preadipocytes from SAT of WT and *Angpt2^{iΔAd}* mice and stimulated them with adipogenic cocktail. Then, primary cultured adipocytes were measured for O₂ consumption using Seahorse XFe96 analyzer. Again, we could not examine any significant changes in among both samples (Supplemental Fig. 3b-e). We included these additional findings and their descriptions into the revised manuscript accordingly as below.

Supplemental Fig 3

a Immunoblot analysis of indicated proteins in SAT of WT and *Angpt2* ^{Δ Ad} mice. BAT is used as a positive control. **b-e** Comparisons of basal respiration, maximal respiration, proton leak, and sparse respiratory capacity in primary cultured adipocytes isolated from SAT in WT and *Angpt2* ^{Δ Ad} mice. Each dot indicates a value obtained from one sample; n = 6/group. Horizontal bars indicate mean \pm SD and P values versus WT by two-tailed Student's t-test. NS, not significant.

Results (Page 5)

However, these smaller SAT adipocytes in the *Angpt2* ^{Δ Ad} mice did not show any signs of apoptosis, beiging, oxygen consumption, immune cell infiltration, or defective vascularization (Fig. 1f-k and Supplementary Fig. 3a-g)...

Methods (Page 18)

Oxygen consumption rate

The oxygen consumption rate was measured using the Seahorse XFe96 analyzer (Seahorse Bioscience) following the manufacturer's instructions. Briefly, ECAR and OCR were measured after primary cultured preadipocytes were stimulated with adipogenic cocktail on XFe96 microplates. Cells were maintained in non-buffered assay medium in a non-CO₂ incubator for 1 hour before the assay. The Mito stress test kit (Seahorse Bioscience) was used to test the OCR under basal conditions in the presence of oligomycin (1.5 μ M), the mitochondrial uncoupler carbonyl cyanide-4-(trifluoromethoxy)phenylhydrazone (FCCP; 1 μ M), and the respiratory chain inhibitors rotenone and antimycin A (0.5 μ M).

Comment 9: Some of the observed changes could also be achieved due to altered immune cell infiltration. As *Angpt2* is involved in this process, the authors should determine myeloid and lymphocyte infiltration rates in the various adipose tissue depots.

Response: This is a valid point that needs to be addressed. To address this issue, we performed additional experiments to explore immune cell infiltration into adipose tissue depots of *Angpt2*^{iΔAd} mice. Of note, there were no differences in the ratio of F4/80⁺ or CD11b⁺ immune cells and CD11c⁺ dendritic cells in both SAT and VAT of WT and *Angpt2*^{iΔAd} mice fed with normal chow diet (Fig. 1 and Supplemental Fig. 3f,g). Thus, adipocyte-derived *Angpt2* itself is not involved in immune cell infiltration as seen in naïve condition. We included these additional results and their descriptions into the revised manuscript accordingly as below.

Supplemental Fig. 3

f, g Representative images and comparisons of indicated immune cell infiltration in SAT and VAT of WT and *Angpt2*^{iΔAd} mice. Magnified view is shown in right panels. Each dot indicates a value obtained from one mouse; *n* = 3 mice/group pooled from two independent experiments. Horizontal bars indicate mean ± SD and *P* values versus WT by two-tailed Student's t-test. NS, not significant. Scale bars, 30 μm.

Comment 10: Fig 6 C: quantification of microvessel density is missing.

Response: We analyzed vascular density with our established method (Park et al., *Nat Commun.*, 2017; Kim et al., *J Clin Invest.*, 2017) in *ITGB1*^{iΔEC} mice. Of note, we still detected no difference in vessel density and area between WT and *ITGB1*^{iΔEC} mice (Supplemental Fig. 9d), which signifies the changes in trans-endothelial FA

uptake are not affected by vascularity itself. We included these additional findings and their descriptions into the revised manuscript accordingly as below.

Supplemental Fig.9

d Representative images and comparison of vascular density and CD31⁺ vessel area in SAT of WT and *ITGβ1*^{ΔEC} mice. Each dot indicates a value obtained from one mouse; *n* = 3 mice/group pooled from two independent experiments. Horizontal bars indicate mean ± SD and *P* values versus WT by two-tailed Student's *t*-test. NS, not significant. Scale bars, 70 μm

Results (Page 9)

There were no changes in vascular integrity or density in *ITGβ1*^{ΔEC} mice after 1 week of tamoxifen treatment (Supplementary Fig. 9a-d).

*References for responses to comments of Reviewer #2

Jang C. et al. (2016). A branched-chain amino acid metabolite drives vascular fatty acid transport and causes insulin resistance. *Apr*;22(4):421-6.

Khalifeh-Soltani, A. et al. (2014). Mfge8 promotes obesity by mediating the uptake of dietary fats and serum fatty acids. *Nat Med*. 20, 175-183.

Strilić B et al. (2009). The molecular basis of vascular lumen formation in the developing mouse aorta. *Dev Cell*. Oct;17(4):505-15.

Kim J et al., (2017). YAP/TAZ regulates sprouting angiogenesis and vascular barrier maturation. *J Clin Invest*. Sep 1;127(9):3441-3461.

Nam M et al., (2017). Mitochondrial retrograde signaling connects respiratory capacity to thermogenic gene expression. *Sci Rep*. May 17;7(1):2013.

Park DY et al., (2017). Plastic roles of pericytes in the blood-retinal barrier. *Nat Commun*. May 16;8:15296.

Pitter B, Werner AC, Montanez E. (2018). Parvins Are Required for Endothelial Cell-Cell Junctions and Cell Polarity During Embryonic Blood Vessel Formation. *Arterioscler Thromb Vasc Biol*. May;38(5):1147-1158.

Reviewer #3:

(Remarks to the Author)

The paper by Bae et al. provides substantial experimental evidence that the angiotensin-2–integrin $\alpha 5\beta 1$ pathway is distinctively involved in a complex crosstalk between adipocytes and endothelial cells of subcutaneous fat, allowing delivery and storage of circulating fatty acids to subcutaneous adipocytes. Inadequate lipid storing abilities of subcutaneous fat is now regarded as a major pathophysiological mechanism leading over time in obese patients to high circulating lipid levels, abnormal lipid storage in target organs, cell lipotoxicity and onset of several metabolic complications of obesity, in particular insulin resistance and diabetes. Results are novel and, in view of the current wide medical interest for obesity and related diseases, the present findings will be of high interest for a wide audience of scholars. In my opinion, the experimental approaches involving both in vitro studies and animal models are well conceived and the methods are adequate and sufficiently detailed to allow a researcher to reproduce the experiments. Statistical analyses are appropriate. I only have one major concern and some minor suggestions to improve the paper.

Response: We appreciate these favorable comments. We hope that the reviewer would be satisfied with our responses to the following specific issues.

Comment 1: The phenotypical characterization of *angpt2^{iΔAd}* mice challenged with high-fat diet is a bit scant. Given the “whitened” morphological aspect of interscapular brown fat, the authors should measure the expression of UCP1 and other thermogenic markers in this depot and compare the results obtained with the respective controls. In addition, some obesity-linked inflammatory markers should be measured in the visceral depots to substantiate this model of obesity. Finally, evaluation of at least leptin and adiponectin blood levels in these mice could offer additional information to the dysmetabolic condition and insulin resistance present in these mice.

This is a valid point that needs further explanation, and please note that a part of this issue was also discussed in Reviewer 2’s comment 7. As the reviewer requested, we performed additional experiments to examine the changes in *Angpt2^{iΔAd}* mice induced by high fat diet. We first examined changes in thermogenic markers and mitochondrial morphology in BAT of *Angpt2^{iΔAd}* mice. Of note, thermogenic markers such as *Ucp1*, *Dio2*, *Pgc1 α* , *Cidea*, and *Elovl3* were significantly decreased in BAT of *Angpt2^{iΔAd}* mice (Supplementary Fig. 12c). Moreover, cristae of mitochondria were unpacked and filled with vacuoles (Supplementary Fig. 12d), altogether indicating defects in thermogenesis (Nam M et al., *Sci Rep.*, 2017). Next, we examined obesity-linked inflammatory markers in VAT of *Angpt2^{iΔAd}* mice. Importantly, inflammatory markers such as *F4/80*, *Cd11b*, *Il-6*, *Rantes*, and *Mcp1* were significantly increased in VAT of *Angpt2^{iΔAd}* mice challenged under high fat diet (Supplementary Fig. 12b). Finally, we evaluated blood levels of leptin and adiponectin in *Angpt2^{iΔAd}* mice. While plasma level of adiponectin was significantly decreased, leptin was elevated in *Angpt2^{iΔAd}* mice (Supplementary Fig. 12e,f).

Altogether, these results substantiate the dysmetabolic condition of *Angpt2*^{iΔAd} mice during high fat diet challenge. We included these additional results and their descriptions into the revised manuscript accordingly as below.

Supplemental Figure 12

b Comparisons of indicated mRNA expression in VAT between WT and *Angpt2*^{iΔAd} mice.

Each dot indicates a value obtained from one mouse; n = 5-6 mice/group. Horizontal bars indicate mean ± SD and P values versus WT by two-tailed Student's t-test.

c Comparisons of indicated mRNA expression in BAT between WT and *Angpt2*^{iΔAd} mice.

Each dot indicates a value obtained from one mouse; n = 5-6 mice/group. Horizontal bars indicate mean ± SD and P values versus WT by two-tailed Student's t-test.

d Representative electron microspore images indicating mitochondrial morphology in BAT of WT and *Angpt2*^{iΔAd} mice. Magnified view is shown in right panels. Scale bars, 2 μm.

e, f Comparison of plasma adiponectin and leptin levels between WT and *Angpt2*^{iΔAd} mice. Each dot indicates a value obtained from one mouse; n = 8 mice/group pooled from two independent experiments. Horizontal bars indicate mean ± SD and P values versus WT by two-tailed Student's t-test. NS, not significant

Results (Page 11)

Obesity-associated inflammatory markers in VAT were upregulated, while thermogenic markers were downregulated in BAT in *Angpt2^{iΔAd}* mice (Supplementary Fig. 12b,c). Likewise, electron microscope analysis revealed unpacked cristae and vacuole-filled mitochondria in BAT of *Angpt2^{iΔAd}* mice (Supplementary Fig. 12d). Moreover, circulating triglyceride and leptin levels were each ~1.5-fold and ~2.0-fold higher, whereas plasma adiponectin level was 30% less in *Angpt2^{iΔAd}* mice (Fig. 9c and Supplementary Fig. 12e,f).

Comment 2: Please, specify what you mean for visceral adipose tissue in mice. Is it the epididymal fat depot?

Response: We have now included the details about the specific adipose depots used in our experiments in the revised Methods as shown in the example below.

Methods (Page 16)

For sampling of tissues, inguinal white adipose tissue was used for subcutaneous adipose tissue (SAT), epididymal white adipose tissue was used for visceral adipose tissue (VAT), interscapular brown adipose tissue (BAT), and quadriceps (skeletal muscle) were used for this study.

Comment 3: The brown adipose tissue depot you analyzed is the interscapular (not “intrascapular”).

Response: Thank you for pointing out this error. We now corrected the description in the revised Methods as shown in the example below.

Methods (Page 16)

For sampling of tissues, inguinal white adipose tissue was used for subcutaneous adipose tissue (SAT), epididymal white adipose tissue was used for visceral adipose tissue (VAT), ~~intrascapular~~ interscapular brown adipose tissue (BAT), and quadriceps (skeletal muscle) were used for this study.

Comment 4: Even whether brown fat expresses much less angiopoietin-2 than subcutaneous white adipose tissue it should be clearly stated (and discussed) that the use of adiponectin-Cre mice involves knocking out angiopoietin-2 not only in white adipocytes but also in brown adipocytes.

Response: This is a valid point. In fact, the *Adiponectin-CreERT2* we used in this study labels 97~99% of white adipocytes, while it labels less than 15% of brown adipocytes (Sassman et al., *Genesis*, 2010). Thus, we believe the effects of white adipocyte-driven *Angpt2* are responsible in the changes we observed in this study. Nevertheless, we agree with the limitation that depletion of *Angpt2* would also affect the expression in brown adipocytes, and therefore we mentioned this limitation in the revised manuscript as below.

Discussion (Page 14)

...In order to do so, manipulating Angpt2 in a depot-specific manner could better define the role of Angpt2 in various adipose depots.

Comment 5: In Fig. 3f-h is not clear what is obtained by immunohistochemistry and what is obtained by using eGFP expressing mice.

Response: We appreciate this comment. Our intention was to highlight the specific expression of ITG α 5 β 1 receptor in endothelial cells of SAT as shown in RNA-Seq (Fig. 3e). To emphasize our intention, we included the expression of ITG α 5 β 1 and Tie2 (by using Tie2-eGFP mice) in other fat depots which clearly show preferential expression of ITG α 5 β 1, but not Tie2, in endothelial cells of SAT (Fig. 3f,g). We included these additional results and their descriptions into the revised manuscript accordingly as below.

Results (Page 7)

Immunohistochemistry analyses supported these reciprocal expression levels of integrin α 5 β 1 and Tie2 in the ECs of SAT, which clearly differs with other depots (Fig. 3f, g and Supplementary Fig. 5a, b).

Comment 6: In Supplementary Figure 3 b liver pics are not comparable (one shows hepatocytes around a portal space, the other shows hepatocytes around a centrolobular vein)

Response: Thank you for pointing out this error. Following the reviewer's comment, we replaced Revised Supplementary Fig. 9f with comparable liver images as below.

Supplementary Fig. 9

Representative H&E-stained images of indicated organs of WT and *ITGβ1*^{ΔEC} mice. Scale bars, 50 μm.

Comment 7: Please, explain in a few words what the vascular permeability test is.

Response: We appreciate this comment. We included detailed information regarding the vascular permeability test in the revised manuscript as shown in the example below.

Results (Page 7)

We found no difference in vascular leakage by trans-well endothelial layer permeability assay, indicating that *Angpt2* stimulates FA uptake independently of vascular permeabilizing actions (Fig. 4f).

Methods (Page 23)

In vitro vascular permeability assay

For assessment of endothelial layer permeability, we modified the manufacturer's instructions of in vitro vascular permeability assay (Millipore). Briefly, HUVECs were grown until confluence for 2-3 days on transwell inserts. FITC-dextran solution (Millipore) was added to the upper chamber and transferred solution was measured.

Comment 8: Data from humans on the expression of angiopoietin-2 in non-diabetic obese patients are interesting; however, a reader also wonders what happens in diabetic obese patients. Are some data available?

Response: In fact, we provided expression values of diabetic obese patients in Supplementary Table 2. We compared the values of non-diabetic obese patients to diabetic obese patients, and could analyze that *Angpt2* was enriched in SAT of non-diabetic obese patients. To clarify, we included detailed information regarding human datasets in the revised methods.

Methods (Page 24)

Comparison of non-diabetic obese versus diabetic obese individuals

...Datasets including GEO reference number, sample number, organ, normalized Angpt2 expression, and status of patient (diabetic, obese) are indicated in Supplementary Table 2...

Comment 9: Please, specify in simple words what Weibel–Palade bodies are.

Response: We have now included more detailed information about Weibel-Palade bodies in the revised manuscript as below.

Discussion (Page 13)

Angpt2 is stored in repository granules of ECs called Weibel–Palade bodies, and rapidly released upon stimulation as an angiocrine factor^{44,45}.

Comment 10: I would expect the authors to discuss their results also in view of the well-known clinical observation that pear-shaped obesity that is more frequent in females and linked to enlargement of some subcutaneous depots is less morbid than the apple-shaped obesity that is more frequent in males and due to fat accumulation in visceral depots.

Response: As we addressed in comment 5 for Reviewer 1, we were not able to examine significant differences in adipose tissue weight and size of female *Angpt2*^{ΔAD} mice compared with male *Angpt2*^{ΔAD} mice (Reviewer only Fig 2). Therefore, we suspect that Angpt2-mediated endothelial FA uptake is not a differential factor in sexual dimorphism of body fat distribution during naïve condition. However, as the reviewer mentioned, its role in obesity is unclear. We have discussed about this important view raised by the reviewer in the revised manuscript as below.

Discussion (Page 14)

Meanwhile, it is well known through clinical observation that subcutaneous obesity is more frequent in females and is less morbid than visceral obesity that is more frequent in males^{2,51}. Whether adipocyte-derived Angpt2 and its role in endothelial fatty acid transport is involved in the sexual dimorphism of body fat distribution needs to be studied.

***References for responses to comments of Reviewer #3**

Nam M et al., (2017). Mitochondrial retrograde signaling connects respiratory capacity to thermogenic gene expression. *Sci Rep.* May 17;7(1):2013.

Sassmann A et al., (2010). Tamoxifen-inducible Cre-mediated recombination in adipocytes. precursors is critical for antiviral immunity. *Genesis.* Oct 1;48(10):618-25.

Reviewers' comments:

Reviewer #1 (Remarks to the Author):

This the study reveals and thoroughly analyzes a novel form of adipocyte-endothelial interaction that controls fatty acid transport and storage in SAT.

The authors have made a comprehensive and detailed response to all previous concerns. The additional data provide substantially more validation of the mouse models used and the conclusions made. Changes to the text provide much more clarity to the experimental methods and data analyses used.

Tara Haas

Reviewer #2 (Remarks to the Author):

The authors have adequately addressed all of my comments. The manuscript has been substantially improved. It will contribute significantly to our understanding of organ-specific angiocrine functions.

Reviewer #3 (Remarks to the Author):

I am satisfied with te authors' response to the comments raised. The paper may be accepted for publication.

Antonio Giordano